# 3D-R1: Enhancing Reasoning in 3D VLMs for Unified Scene Understanding

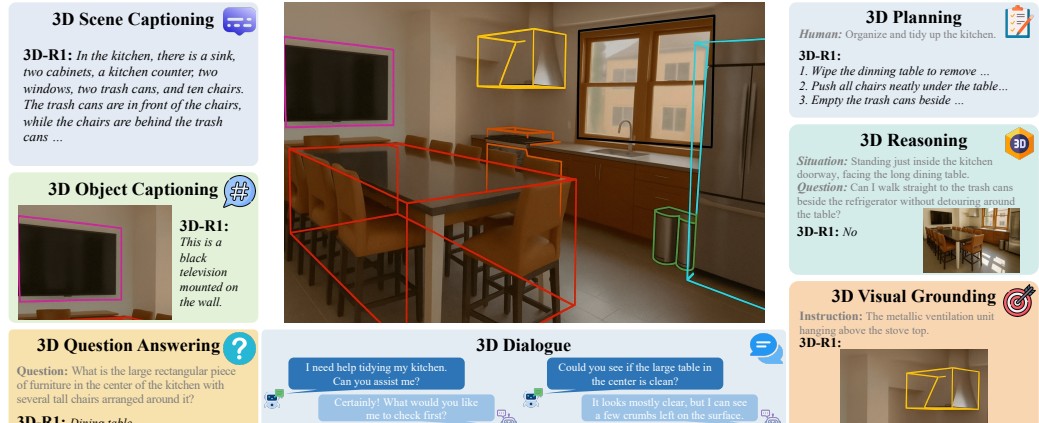

Figure 1: **3D-R1** is an open-source generalist model that enhances the reasoning of 3D VLMs for unified scene understanding.

## Abstract

Large vision-language models (VLMs) have made significant strides in 2D visual understanding tasks, sparking interest in extending these capabilities to 3D scene understanding. However, current 3D VLMs often struggle with robust reasoning and generalization due to limitations in high-quality spatial data and the static nature of viewpoint assumptions. To address these challenges, we propose **3D-R1**, a foundation model that enhances the reasoning capabilities of 3D VLMs. Specifically, we first construct a high-quality synthetic dataset with CoT, named Scene-30K, leveraging existing 3D-VL datasets and a data engine based on Gemini 2.5 Pro. It serves as cold-start initialization data for 3D-R1. Moreover, we leverage RLHF policy such as GRPO in the reinforcement learning training process to enhance reasoning capabilities and introduce three reward functions: a perception reward, a semantic similarity reward and a format reward to maintain detection accuracy and answer semantic precision. Furthermore, we introduce a dynamic view selection strategy that adaptively chooses the most informative perspectives for 3D scene understanding. Extensive experiments demonstrate that 3D-R1 delivers an average improvement of 10% across various 3D scene benchmarks, highlighting its effectiveness in enhancing reasoning and generalization in 3D scene understanding.

## 1 Introduction

3D scene understanding is a fundamental capability for intelligent systems, enabling a wide range of applications in embodied AI, robotics, and mixed reality (Zhao et al., 2024; Song et al., 2025). The ability of an agent to perceive and reason about 3D environments is crucial for tasks such as robotic manipulation, navigation, and long-horizon planning. Similarly, context-aware augmented and virtual reality applications require a rich semantic understanding of physical spaces to anchor virtual content and interactions in the real world. Furthermore, 3D scene understanding facilitates

Table 1: **Statistics of the public 3D-VL datasets that we draw on when synthesising the Scene-30K dataset.** "3D Scene / Obj." give the number of reconstructed scenes and annotated objects respectively. "Task" indicates the original benchmark focus, "DC" stands for Dense Captioning, "QA" for Question Answering, "VG" for Visual Grounding, and "MT" for Multi-tasking. "Anno." denotes language from human annotations and "Syn." for template-based or LLM generated descriptions.

| Dataset | 3D Scene | Obj. | Task | Obj. Caption | Scene Caption | Obj. Referral | Quality Check | Anno. | Syn. | Total |
|---------|----------|------|------|-------------|---------------|---------------|---------------|-------|------|-------|
| ScanRefer Chen et al. (2020) | 800 | - | DC&VG | ✗ | ✗ | ✓ | ✓ | 52K | - | 52K |
| Nr3D Achlioptas et al. (2020) | 707 | - | DC&VG | ✗ | ✗ | ✓ | ✓ | 42K | 200K | 242K |
| ScanQA Azuma et al. (2022) | 1.5K | 33K | QA | - | - | - | ✓ | 27K | - | 27K |
| SceneVerse Jia et al. (2024) | 68K | 1.5M | DC&VG | ✓ | ✓ | ✓ | ✓ | 190K | 2.3M | 2.5M |
| **Scene-30K (Ours)** | 1.5K | 33K | MT | ✓ | ✓ | ✓ | ✓ | - | 30K | 30K |

advanced spatial reasoning, such as interpreting spatial relations or inferring hidden object configurations, essential for agents to interact naturally with complex environments.

Researchers have recently extended vision-language models into the 3D domain to tackle tasks like 3D scene dense captioning (3D-DC) (Chen et al., 2021b; 2023b; 2024b), 3D object captioning (Luo et al., 2024), 3D question answering (3D-QA) (Azuma et al., 2022; Mo & Liu, 2024), 3D dialogue (Chen et al., 2024a; Halacheva et al., 2025), 3D visual grounding (3D-VG) (Jia et al., 2024; Huang et al., 2024a), and 3D reasoning and planning (Halacheva et al., 2025; Chen et al., 2024a), as shown in Figure 1.

Despite this progress, current 3D vision language models still face significant limitations. One of the primary challenges is enabling models to reason about complex spatial relationships and dynamic scene contexts. Traditional supervised fine-tuning (SFT) approaches often fail to effectively generalize across varied environments, as they are limited by the static nature of their training data and lack of adaptability. Another limitation is the reliance on pre-defined views or representations. Several pipelines assume a fixed set of camera viewpoints or a global panoramic scene encoding, which can introduce irrelevant visual content and still miss critical details occluded in those views.

Recently, DeepSeek-R1 (DeepSeek-AI, 2025) has successfully used reinforcement learning (RL) to induce large language models(LLMs) to autonomously emerge complex cognitive reasoning capabilities, begging our thinking to see whether we can leverage reinforcement learning (RL) to improve reasoning ability in 3D VLMs.

To address these challenges, we propose 3D-R1, a foundation model to enhance reasoning capability in 3D scene understanding that integrates cold-start initialization with RL training. First, we synthesize a high-quality 3D scene CoT dataset Scene-30K with diverse question types, as illustrated in Figure 2(b). Specifically, we design a 3D VLM to generate a concise textual description of a scene. This description captures objects, their relations, and their layout. The resulting textual descriptions are then passed to a reasoning model Gemini 2.5 Pro (Team et al., 2025) to produce high-quality CoT reasoning. Finally, the dataset is refined through rule-based data filtering, ultimately obtaining a dataset with 30K complex CoT reasoning samples, which serves as the cold-start initialization dataset for 3D-R1. Building on this foundation, we design a GRPO-based RLHF policy in the reinforcement learning fine-tune process and introduce three reward functions: a format reward, a perception reward, and a semantic similarity reward. This process focuses on enhancing the model's reasoning capabilities while maintaining detection accuracy and answer semantic precision. Furthermore, we introduce a dynamic view selection method, guiding the model learns to assign ranking scores to candidate viewpoints of the 3D scene and dynamically select the most informative views. We conduct extensive experiments to enhance the capacities of reasoning within complex and diverse 3D environments. As shown in Figure 2(c), 3D-R1 achieves strong performance across various 3D scene tasks.

The main contributions of this work are as follows:

- We introduce **3D-R1**, a pioneering 3D VLM that leverages cold-start initialization and RL training to enhance reasoning capability in 3D scene understanding. Specifically, we design RLHF policy based on GRPO, including format, perception and semantic similarity reward function to improve reasoning in complex 3D scenes.

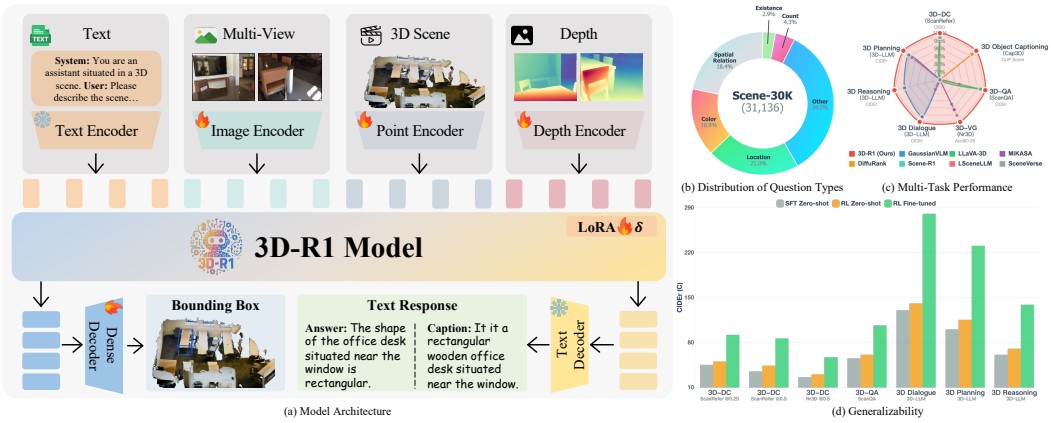

Figure 2: **(a) Architecture.** It takes text, multi-view images, 3D point clouds, and depth maps as input and formulates comprehensive 3D tasks as autoregressive sequence prediction. **(b) Distribution of question types.** Scene-30K contains diverse categories. **(c) Multi-task performance.** 3D-R1 demonstrates strong performance across various tasks. **(d) Generalizability.** 3D-R1 exhibits remarkable generalizability with enhanced reasoning capabilities.

- A high-quality 30K scene CoT dataset is constructed to serve as a cold-start initialization data for 3D VLMs. Furthermore, we introduce dynamic view selection strategy that enables the model to dynamically select views of a 3D scene based on learned relevance scores.
- Extensive experiments demonstrate that 3D-R1 achieves an average improvement of 10% across various 3D scene benchmarks.

## 2 THE PROPOSED METHOD

### 2.1 OVERVIEW

The 3D-R1 framework unfolds in two main phases. In the first phase, we synthesize the Scene-30K dataset, which pairs 3D scenes with questions and coherent chains of thought (CoT). In the second phase, we perform a cold start with the Scene-30K dataset to teach the base 3D VLM shown in Figure 2(a) to reason in a "human-like" fashion. Subsequently, as illustrated in Figure 4 we use RLHF policy such as Group Relative Policy Optimization (GRPO) and introduce two reward functions: a perception reward and a semantic similarity reward during the reinforcement learning training process to enhance the model's reasoning capabilities. Finally, we introduce a dynamic view selection method that scores multiple candidate views of each 3D scene and adaptively chooses the most informative perspectives to answer the questions, ensuring the model focuses on relevant spatial context.

### 2.2 COT DATA ENGINE

We propose a CoT data engine for the construction of Chains of Thought (CoT) (Wei et al., 2022) data tailored to 3D scene understanding. This engine leverages the general reasoning capabilities of the large language model (LLM) to answer the questions with coherent, high-quality Chains of Thought (CoT).

As illustrated in Figure 3, the point cloud of a 3D scene is fed into a scene description generator, which is a pre-trained 3D VLM that produces a concise textual summary of the scene. This summary captures objects, their relations, and their layout. Then we design a comprehensive prompt that instructs Gemini 2.5 Pro (Team et al., 2025) to reason through the detailed logic structure to answer the question from the ScanQA (Azuma et al., 2022) dataset. The prompt provides clear task instructions, specifies the required output format, and includes the previously generated scene description, guiding the model to produce structured step-by-step CoT reasoning. Finally, the model outputs Chains of Thought (CoT) enclosed in <think>...</think> tags, followed by the final answer in <answer>...</answer> tags. By running this pipeline on tens of thousands of 3D

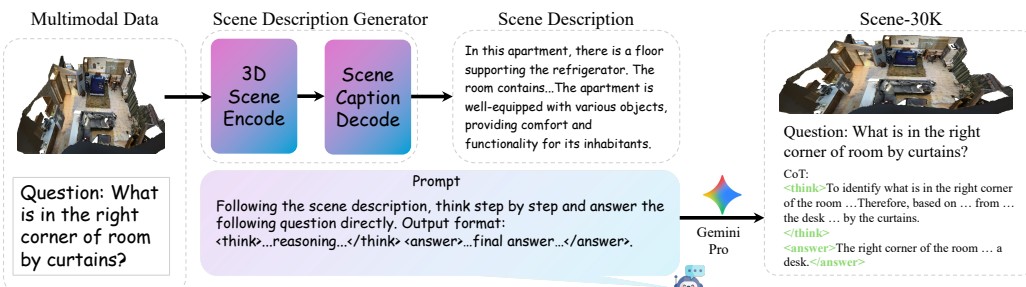

Figure 3: **CoT data engine.** The point cloud of a scene is first sent to scene dscription generator to get a description of the scene. Then based on the description, we apply Gemini 2.5 Pro to synthetic CoT data.

scenes and questions, we collect roughly 35K CoT examples, each containing a scene ID, a question, and the machine-generated `<think>` rationale and `<answer>` output. Then these examples are refined through a rule-based filtering process that eliminates responses with missing structure or inconsistent reasoning; for more details, please see *Appendix* C. Finally, the 30K resulting examples constitute a high-quality CoT reasoning dataset, which we call Scene-30K dataset that serves as the cold-start initialization dataset for 3D-R1.

### 2.3 COLD START STAGE

Inspired by the success of DeepSeek-R1 (DeepSeek-AI, 2025) in solving mathematical reasoning tasks through pure reinforcement learning, we first experiment with end-to-end RL training for our model, with the aim of inducing Chains of Thought (CoT) reasoning to answer the question solely from reward signals. However, this approach proves highly unstable in the 3D VLM base model: the model frequently fails to generate coherent CoT sequences and, more critically, produces answers that are semantically misaligned.

To address the above issues, we adopt a cold start stage based on supervised fine-tuning on the Scene-30K dataset. Leveraging the dataset, containing a question of scene, Chains of Thought (CoT) reasoning process, and corresponding final answer sequences, we fine-tune the 3D vision language model to bootstrap its ability to generate structured outputs in the form `<think>...</think><answer>...</answer>`. This supervised initialization forces the model to learn the expected format for both the multistep reasoning process and the final answer, providing a stable and effective foundation for subsequent policy optimization with reinforcement learning (RL).

### 2.4 REINFORCEMENT LEARNING

GRPO (Shao et al., 2024) introduces an innovative approach rooted in reinforcement learning, showcasing impressive results in models such as DeepSeek R1 (DeepSeek-AI, 2025). Its main objective is to improve the model's reasoning skills by progressively improving its policy, using feedback from the precision of the responses sampled within a group. 3D-R1 decomposes the 3D scene understanding task into two distinct subtasks: scene perception and answer generation. It enables more focused learning and better generalization in complex 3D environments.

**Policy samples.** For a given input state $(x, q)$, where $x$ is the visual encoding of the input point cloud and $q$ is the textual encoding of the question, 3D-R1 first generates $N$ distinct responses $\{o_1, o_1, \cdots, o_N\}$ from the current policy $\pi_\theta$. To better guide policy learning and improve alignment between textual prompts and generated answers, we introduce a multi-reward mechanism.

**Format reward.** To ensure that the content generated by the model has a resolvable structure, we introduce Format Reward $R_{Format}$. This reward detects through regularization expressions whether the generated results strictly follow the predefined format: `<think>Reasoning</think><answer>final answer</answer>`. The Format re-

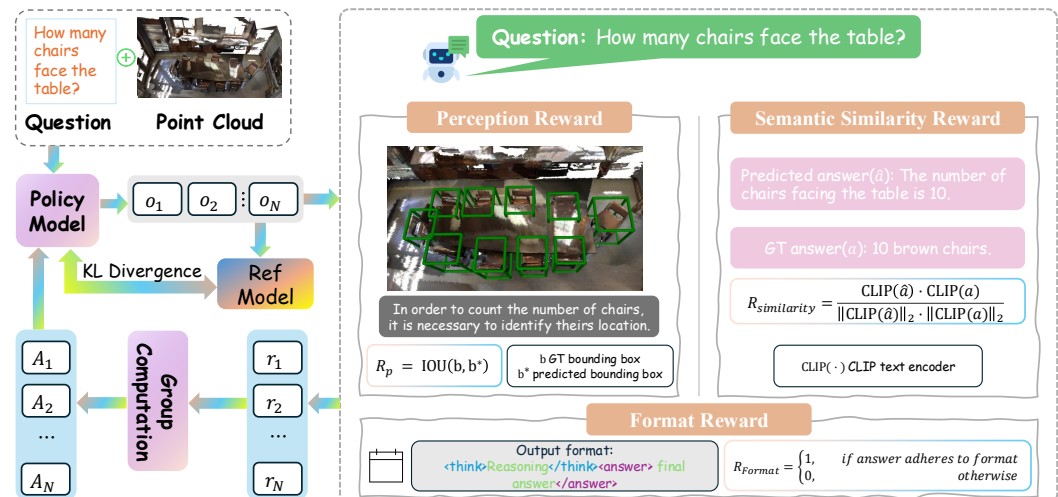

Figure 4: **The pipeline of Reinforcement Learning based GRPO.** The policy model generates $N$ outputs from a point cloud and question. Then perception IoU, semantic CLIP-similarity, and format-adherence rewards are computed, grouped, and combined with a KL term to a frozen reference model to update the policy.

ward is defined as follows:

$$R_{Format} = \begin{cases} 1, & \text{if Answer adheres to format} \\ 0, & \text{otherwise} \end{cases}. \tag{1}$$

**Perception reward.** The perception reward focuses on the core objective of 3D scene perception: accurately identifying where the relevant objects' location is. It evaluates spatial precision by comparing the predicted bounding box $b^*$ with the ground-truth box $b$ using the intersection-over-union (IoU) metric. By optimizing $R_p$, the model is encouraged to generate spatially precise and semantically grounded predictions that directly generate the correct answer. The Perception reward is defined as

$$R_p = \text{IoU}(b, b^*). \tag{2}$$

**Semantic similarity reward.** To encourage semantic coherence between the predicted answer $\hat{a}$ and the ground-truth answer $a$, we adopt a semantic similarity reward $R_{similarity}$. Specifically, we employ a pre-trained text encoder CLIP to obtain feature representations of both answers. The reward is computed as the cosine similarity between their embeddings:

$$R_{similarity} = \frac{\text{CLIP}_{\text{text}}(\hat{a}) \cdot \text{CLIP}_{\text{text}}(a)}{\|\text{CLIP}_{\text{text}}(\hat{a})\|_2 \cdot \|\text{CLIP}_{\text{text}}(a)\|_2}. \tag{3}$$

**Policy update.** Inspired by Group Relative Policy Optimization (GRPO) (Shao et al., 2024), we select multiple responses from the current policy as candidate responses. Each output is assigned a scalar reward, resulting in a reward vector $\mathbf{r} = \{r_1, r_2, \cdots, r_N\}$, computed by task-specific reward functions that evaluate the quality of each output. To assess the quality of each response relative to others, we normalize the rewards by computing the mean and standard deviation:

$$\hat{A}_i = \frac{r_i - \text{mean}(\mathbf{r})}{\text{std}(\mathbf{r})}, \tag{4}$$

where $\hat{A}_i$ denotes the advantage of the $i$-th response. These advantages are then used to update the policy by maximizing the following clipped objective:

$$\mathcal{J}_{\text{GRPO}}(\theta) = \mathbb{E}_c\left[\frac{1}{G}\sum_{i=1}^{G}\left(\min\left(\frac{\pi_\theta(o_i|q)}{\pi_{\theta_{\text{old}}}(o_i|q)}\hat{A}_i, \text{clip}\left(\frac{\pi_\theta(o_i|q)}{\pi_{\theta_{\text{old}}}(o_i|q)}, 1-\varepsilon, 1+\varepsilon\right)\hat{A}_i\right) - \beta\, D_{\text{KL}}(\pi_\theta\|\pi_{\text{ref}})\right)\right]. \tag{5}$$

Table 2: **3D scene dense captioning results on ScanRefer (Chen et al., 2020) and Nr3D (Achlioptas et al., 2020).** For fair comparison, we list methods that are trained under the standard per-word cross-entropy loss without additional 3D scenes. Our proposed 3D-R1 surpasses previous 3D specialists on both datasets.

| Method | ScanRefer | | | | | | | | Nr3D | | | |
|---|---|---|---|---|---|---|---|---|---|---|---|---|
| | C@0.25↑ | B-4@0.25↑ | M@0.25↑ | R@0.25↑ | C@0.5↑ | B-4@0.5↑ | M@0.5↑ | R@0.5↑ | C@0.5↑ | B-4@0.5↑ | M@0.5↑ | R@0.5↑ |
| Scan2Cap Chen et al. (2021b) | 56.82 | 34.18 | 26.29 | 55.27 | 39.08 | 23.32 | 21.97 | 44.78 | 27.47 | 17.24 | 21.80 | 49.06 |
| MORE Jiao et al. (2022) | 62.91 | 36.25 | 26.75 | 56.33 | 40.94 | 22.93 | 21.66 | 44.42 | - | - | - | - |
| SpaCap3D Wang et al. (2022) | - | - | - | - | 44.02 | 25.26 | 22.33 | 45.36 | 33.71 | 19.92 | 22.61 | 50.50 |
| REMAN Mao et al. (2023) | 62.01 | 36.37 | 26.76 | 56.25 | 45.00 | 26.31 | 22.67 | 46.96 | 34.81 | 20.37 | 23.01 | 50.99 |
| D3Net Chen et al. (2021a) | - | - | - | - | 46.07 | 30.29 | 24.35 | 51.67 | 33.85 | 20.70 | 23.13 | 53.38 |
| Contextual Zhong et al. (2022) | - | - | - | - | 46.11 | 25.47 | 22.64 | 45.96 | 35.26 | 20.42 | 22.77 | 50.78 |
| UniT3D Chen et al. (2023a) | - | - | - | - | 46.69 | 27.22 | 21.91 | 45.98 | - | - | - | - |
| 3DJCG Cai et al. (2022) | 64.70 | 40.17 | 27.66 | 59.23 | 49.48 | 31.03 | 24.22 | 50.80 | 38.06 | 22.82 | 23.77 | 52.99 |
| 3D-VLP Jin et al. (2023) | 70.73 | 41.03 | 28.14 | 59.72 | 54.94 | 32.31 | 24.83 | 51.51 | - | - | - | - |
| 3D-VisTA Zhu et al. (2023) | - | - | - | - | 61.60 | 34.10 | 26.80 | 55.00 | - | - | - | - |
| Vote2Cap-DETR Chen et al. (2023b) | 71.45 | 39.34 | 28.25 | 59.33 | 61.81 | 34.46 | 26.22 | 54.40 | 43.84 | 26.68 | 25.41 | 54.43 |
| LL3DA Chen et al. (2024a) | 74.17 | 41.41 | 27.76 | 59.53 | 65.19 | 36.79 | 25.97 | 55.06 | 51.18 | 28.75 | 25.91 | 56.61 |
| Vote2Cap-DETR++ Chen et al. (2024b) | 76.36 | 41.37 | 28.70 | 60.00 | 67.58 | 37.05 | 26.89 | 55.64 | 47.08 | 27.70 | 25.44 | 55.22 |
| LEO Huang et al. (2024b) | - | - | - | - | 72.40 | 38.20 | 27.90 | 58.10 | - | - | - | - |
| ChatScene Huang et al. (2024a) | - | - | - | - | 77.20 | 36.30 | 28.00 | 58.10 | - | - | - | - |
| LLaVA-3D Zhu et al. (2024) | - | - | - | - | 84.10 | 42.60 | 29.00 | 63.40 | - | - | - | - |
| BiCA Kim et al. (2025) | 78.42 | 41.46 | 28.82 | 60.02 | 68.46 | 38.23 | 27.56 | 58.56 | 48.77 | 28.35 | 25.60 | 55.81 |
| 3D CoCa Huang et al. (2025a) | 85.42 | 45.56 | 30.95 | 61.98 | 77.13 | 41.23 | 28.52 | 57.40 | 52.84 | 29.29 | 25.55 | 56.43 |
| 3D-LLaVA Deng et al. (2025) | - | - | - | - | 78.80 | 36.90 | 27.10 | 57.70 | - | - | - | - |
| Spatial 3D-LLM Wang et al. (2025) | - | - | - | - | 72.20 | 34.60 | 23.10 | 54.30 | - | - | - | - |
| **3D-R1 (Ours)** | **91.85** | **48.76** | **32.14** | **62.23** | **86.45** | **44.34** | **29.78** | **64.50** | **56.98** | **31.13** | **26.12** | **57.54** |

## 2.5 DYNAMIC VIEW SELECTION

To bridge the gap between 3D scene representations and the 2D perspective inputs that VLMs expect, we introduce a dynamical view selection module. The core idea is to automatically select a set of informative 2D views from a 3D scene that best convey the content of the scene to the vision-language model.

**Candidate view generation.** For each 3D scene, we first generate a pool of 30 candidate views. We use the 3D point cloud to render RGB images from various viewpoints. In practice, we sample camera positions uniformly around the scene or at strategic locations to obtain a diverse set of perspective images. Each candidate view is processed by a pre-trained visual encoder to extract features. This pre-trained model provides a rich description of the view content without any additional 3D training, capitalizing on the learned 2D visual semantics.

**View scoring metrics.** We design three complementary scoring functions to evaluate each candidate view with respect to a given textual context. These scores are used to prioritize critical and diverse views. Specifically, for each scene $v$ and input text $t$, we calculate $S_{\text{Text}\rightarrow\text{3D}}$, $S_{\text{Image}\rightarrow\text{3D}}$, and $S_{\text{CLIP}}$ as follows:

$$S_{\text{Text}\rightarrow\text{3D}}(v,t) = \frac{E_{\text{text}}(t) \cdot E_{\text{3D}}(v)}{\|E_{\text{text}}(t)\|_2 \|E_{\text{3D}}(v)\|_2},$$

$$S_{\text{Image}\rightarrow\text{3D}}(v,t) = \frac{1}{|I(t)|} \sum_{i \in I(t)} \frac{E_{\text{img}}(i) \cdot E_{\text{3D}}(v)}{\|E_{\text{img}}(i)\| \|E_{\text{3D}}(v)\|}, \tag{6}$$

$$S_{\text{CLIP}}(v,t) = \frac{E_{\text{CLIP}}^{\text{txt}}(t) \cdot E_{\text{CLIP}}^{\text{img}}(R(v))}{\|E_{\text{CLIP}}^{\text{txt}}(t)\| \left\|E_{\text{CLIP}}^{\text{img}}(R(v))\right\|},$$

where $E_{\text{text}}(\cdot)$ denotes text encoder, $E_{\text{img}}(\cdot)$ denotes image encoder, $E_{\text{3D}}(\cdot)$ denotes point encoder, $I(t)$ is the set of multi-view images of the scene, $R(v)$ renders scene $v$ into 2D image, $E_{\text{CLIP}}^{\text{txt}}(\cdot)$ and $E_{\text{CLIP}}^{\text{img}}(\cdot)$ are the text and image branches of CLIP, and $\|\cdot\|$ is the Euclidean norm.

**Dynamic score fusion.** The above scores are combined to produce an overall utility score for each view $U(v)$. Instead of manually tuning their relative importance, we dynamically learn the weight of these components. We introduce learnable parameters $w_t, w_c, w_{clip}$ for the text relevance, coverage, and CLIP alignment scores, respectively. This adaptive fusion ensures that $U(v)$ emphasizes the most useful views for each scenario. $U(v)$ is defined as follows:

$$U(v) = w_t \cdot S_{\text{Text}\rightarrow\text{3D}} + w_c \cdot S_{\text{Image}\rightarrow\text{3D}} + w_{clip} \cdot S_{\text{CLIP}}, \tag{7}$$

where $w_c + w_{clip} = 1$, $w_t$ as an independent scalar. This allows the model to dynamically adjust the influence of textual grounding relative to visual signals. To stabilize training, we apply an L2

Table 3: **3D question answering results on ScanQA Azuma et al. (2022).** 3D-R1 out-performs previous methods on the validation set and two test sets.

| Method | Validation | | | | Test w/ object | | | | Test w/o object | | | |
|---|---|---|---|---|---|---|---|---|---|---|---|---|
| | C↑ | B-4↑ | M↑ | R↑ | C↑ | B-4↑ | M↑ | R↑ | C↑ | B-4↑ | M↑ | R↑ |
| ScanQA Azuma et al. (2022) | 64.86 | 10.08 | 13.14 | 33.33 | 67.29 | 12.04 | 13.55 | 34.34 | 60.24 | 10.75 | 12.59 | 31.09 |
| Clip-Guided Parelli et al. (2023) | - | - | - | - | 69.53 | 14.64 | 13.94 | 35.15 | 62.83 | 11.73 | 13.28 | 32.41 |
| 3D-VLP Jin et al. (2023) | 66.97 | 11.15 | 13.53 | 34.51 | 70.18 | 11.23 | 14.16 | 35.97 | 63.40 | 15.84 | 13.13 | 31.79 |
| 3D-LLM Hong et al. (2023) | 69.40 | 12.00 | 14.50 | 35.70 | 69.60 | 11.60 | 14.90 | 35.30 | - | - | - | - |
| 3D-VisTA Zhu et al. (2023) | 69.60 | 10.40 | 13.90 | 35.70 | 68.60 | 10.50 | 13.80 | 35.50 | 55.70 | 8.70 | 11.69 | 29.60 |
| LL3DA Chen et al. (2024a) | 76.79 | 13.53 | 15.88 | 37.31 | 78.16 | 13.97 | 16.38 | 38.15 | 70.29 | 12.19 | 14.85 | 35.17 |
| BridgeQA Mo & Liu (2024) | - | - | - | - | 83.75 | 24.06 | 16.51 | 43.26 | 79.34 | 17.74 | 15.60 | 41.18 |
| ChatScene Huang et al. (2024a) | 87.70 | 14.30 | 18.00 | 41.60 | - | - | - | - | - | - | - | - |
| 3D-LLaVA Deng et al. (2025) | 92.60 | 17.10 | 18.40 | 43.10 | - | - | - | - | - | - | - | - |
| Scene-LLM Fu et al. (2025) | 80.00 | 12.00 | 16.60 | 40.00 | - | - | - | - | - | - | - | - |
| Spatial 3D-LLM Wang et al. (2025) | 82.50 | 13.90 | 16.80 | 39.10 | - | - | - | - | - | - | - | - |
| LSceneLLM Zhi et al. (2025) | 88.24 | - | 17.95 | 40.82 | - | - | - | - | - | - | - | - |
| LEO Huang et al. (2024b) | 101.40 | 13.20 | 20.00 | 49.20 | - | - | - | - | - | - | - | - |
| LLaVA-3D Zhu et al. (2024) | 103.10 | 16.40 | 20.80 | 49.60 | - | - | - | - | - | - | - | - |
| GaussianVLM Halacheva et al. (2025) | - | - | **22.90** | 34.80 | - | - | - | - | - | - | - | - |
| **3D-R1 (Ours)** | **106.45** | **17.80** | 22.13 | **51.23** | **94.65** | **35.34** | **27.34** | **54.35** | **89.56** | **26.34** | **27.34** | **52.38** |

Table 4: **3D object captioning results** on Cap3D (Luo et al., 2023). † indicates DiffuRank (Luo et al., 2024) trained with top 6 views.

| Method | Quality A/B test | | | Hallucination A/B test | | | CLIP | | | |
|---|---|---|---|---|---|---|---|---|---|---|
| | Score(1-5) | Win % | Lose % | Score(1-5) | Win % | Lose % | Score | R@1 | R@5 | R@10 |
| Cap3D Luo et al. (2023) | 2.62 | 32.70 | 60.20 | 2.43 | 25.80 | 63.90 | 71.20 | 20.50 | 40.80 | 51.90 |
| DiffuRank (Allviews 28-views) | 2.91 | 37.90 | 43.60 | 2.85 | 35.10 | 47.20 | 73.50 | 24.90 | 46.70 | 55.70 |
| DiffuRank (Horizontal 6-views) | 2.84 | 35.20 | 44.50 | 2.90 | 36.20 | 40.90 | 73.80 | 25.80 | 46.70 | 55.90 |
| DiffuRank (Bottom 6-views) | 2.74 | 31.10 | 52.00 | 2.61 | 30.10 | 57.00 | 72.80 | 4.60 | 45.10 | 55.20 |
| DiffuRank Luo et al. (2024)† | - | - | - | - | - | - | 74.60 | 26.70 | 48.20 | 57.50 |
| **3D-R1 (Ours)** | **4.32** | **34.56** | **65.34** | **4.21** | **27.34** | **69.12** | **77.34** | **32.23** | **55.45** | **63.12** |

regularization term on $w_t$, encouraging it to stay near a target value (e.g., $\mu = 0.3$), which prevents overly dominant text influence.

# 3 EXPERIMENTS

## 3.1 DATASETS AND METRICS

**Datasets.** To obtain the cold-start dataset, as shown in Tab 1, we use ScanQA (Azuma et al., 2022), ScanRefer (Chen et al., 2020), Nr3D (Achlioptas et al., 2020) and SceneVerse (Jia et al., 2024) datasets to synthesize the Scene-30K dataset. In downstream tasks, we incorporate standard benchmarks including ScanRefer (Chen et al., 2020) and Nr3D (Achlioptas et al., 2020) dataset for 3D-DC and 3D-VG, Cap3D (Luo et al., 2023) for 3D object captioning, ScanQA (Azuma et al., 2022) dataset for 3D-QA , 3D-LLM (Hong et al., 2023) for 3D dialogue and planning and SQA3D (Ma et al., 2023) for 3D reasoning.

**Metrics.** For 3D-DC, 3D-QA, 3D dialogue, 3D reasoning and 3D planning tasks, we use the metrics CIDEr (Vedantam et al., 2015), BLEU (Papineni et al., 2002), METEOR (Banerjee & Lavie, 2005) and ROUGE-L (Lin, 2004), which are briefly denoted by C, B-4, M and R, respectively, to evaluate the quality of the generated textual responses. For 3D-VG task, we use metric Acc@$s$IoU, which reports grounding accuracy with different IoU scores $s$ between the predicted and ground truth bounding boxes. For the 3D object captioning task, we adopt both human and automated evaluation metrics. Human evaluation involves A/B testing to assess two key aspects: caption quality and hallucination rate, reporting average preference scores and win/loss rates. For automated evaluation, we follow CLIP-based retrieval metrics, including cosine similarity scores and retrieval precision (Poole et al., 2023) at the top-1, top-5 and top-10 (R@1, R@5, R@10).

## 3.2 IMPLEMENTATIONS DETAILS

**Architecture.** We construct the encoder and decoder components on top of the base VLM, Qwen2.5-VL-7B-Instruct (Bai et al., 2025). We adopt SigLIP-2 (ViT-L/14) (Tschannen et al., 2025), Depth-Anything v2 (ViT-L/14) (Yang et al., 2024), and Point Transformer v3 (Wu et al., 2024) as image, depth and point cloud encoders, respectively. The output from each encoder is linearly pro-

Table 5: **3D dialogue and planning** results on 3D-LLM (Hong et al., 2023). **3D reasoning** results on SQA3D (Ma et al., 2023).

| Method | Dialogue | | | | Reasoning | | | | Planning | | | |
|---|---|---|---|---|---|---|---|---|---|---|---|---|
| | C↑ | B-4↑ | M↑ | R↑ | C↑ | B-4↑ | M↑ | R↑ | C↑ | B-4↑ | M↑ | R↑ |
| LL3DA Chen et al. (2024a) | 190.01 | 23.95 | 23.50 | 40.61 | - | - | - | - | 128.80 | 12.95 | 17.05 | 39.25 |
| Spatial 3D-LLM Wang et al. (2025) | - | - | - | - | - | - | - | - | 195.92 | 14.65 | 18.95 | 36.93 |
| LSceneLLM Zhi et al. (2024) | 104.98 | - | 21.26 | 36.00 | - | - | - | - | 214.63 | - | 21.05 | 47.05 |
| LEO Huang et al. (2024b) | - | - | - | - | 124.70 | 9.40 | 25.50 | 48.40 | - | - | - | - |
| GPT-4o OpenAI (2024) | 200.34 | 26.47 | 26.35 | 47.88 | 120.45 | 19.34 | 25.45 | 49.34 | 210.23 | 18.67 | 42.23 | 45.23 |
| Gemini 2.5 Pro Team et al. (2025) | 210.23 | 27.34 | 28.12 | 48.22 | 125.23 | 20.23 | 27.34 | 55.34 | 215.34 | 20.19 | 44.34 | 46.23 |
| GaussianVLM Halacheva et al. (2025) | 270.10 | 31.50 | 55.70 | 48.60 | 129.60 | 17.10 | 26.40 | 50.20 | 220.40 | 20.30 | 44.50 | 48.00 |
| **3D-R1 (Ours)** | **280.34** | **39.45** | **66.89** | **55.34** | **138.67** | **23.56** | **35.45** | **60.02** | **230.50** | **25.45** | **48.34** | **55.67** |

Table 6: **3D visual grounding** results on ScanRefer (Chen et al., 2020) and Nr3D (Achlioptas et al., 2020).

| Method | Nr3D | ScanRefer | |
|---|---|---|---|
| | Acc@0.25 | Acc@0.5 | Acc@0.25 |
| 3DVG-Trans Lichen et al. (2021) | 40.80 | 34.70 | 47.60 |
| TGNN Huang et al. (2021) | 37.30 | 29.70 | 37.37 |
| TransRefer3D He et al. (2021) | 48.00 | - | - |
| InstanceRefer Yuan et al. (2021) | 38.80 | 32.93 | 40.23 |
| FFL-3DOG Feng et al. (2021) | 41.70 | 34.01 | 41.33 |
| LAR BAKR et al. (2022) | 48.90 | - | - |
| SAT Yang et al. (2021) | 56.50 | 30.14 | 44.54 |
| 3D-SPS Luo et al. (2022) | 51.50 | 36.98 | 48.82 |
| 3DJCG Cai et al. (2022) | - | 37.33 | 49.56 |
| BUTD-DETR Jain et al. (2022) | 54.60 | 39.80 | 52.20 |
| MVT Huang et al. (2022) | 59.50 | 33.26 | 40.80 |
| ViL3DRel Chen et al. (2022) | 64.40 | 37.73 | 47.94 |
| EDA Wu et al. (2023) | 52.10 | 42.26 | 54.59 |
| 3D-VisTA Zhu et al. (2023) | 64.20 | 45.80 | 50.60 |
| SceneVerse Jia et al. (2024) | 64.90 | 48.10 | - |
| ChatScene Huang et al. (2024a) | - | 50.20 | 55.50 |
| LLaVA-3D Zhu et al. (2024) | - | 42.70 | 50.10 |
| Video-3D LLM Zheng et al. (2025) | - | 51.72 | 58.12 |
| GPT4Scene Qi et al. (2025) | - | 57.00 | 62.60 |
| MiKASA Chang et al. (2024) | 64.40 | - | - |
| Scene-R1 Yuan et al. (2025) | - | 17.10 | 38.80 |
| **3D-R1 (Ours)** | **68.80** | **59.24** | **65.85** |

jected to match the dimensionality of the text tokens and concatenated with them to form a unified sequence. And we freeze the entire backbone, including the text encoder and decoder, and fine-tune only the 12-layer LoRA adapters, the image encoder, the point cloud encoder, the depth encoder, and the dense decoder.

**Parameter efficient tuning.** To enable efficient fine-tuning, we inject LoRA adapters (Hu et al., 2022) into the last 8 transformer blocks of the base VLM, which comprises 28 transformer blocks. In each selected block, LoRA is implemented for all projection matrices in the VLM, *i.e.*, $(W_q, W_k, W_v, W_o)$ in attention modules and $(W_{gate}, W_{up}, W_{down})$ in MLPs. Each adapter is configured with rank $\delta = 12$, scaling factor $\alpha = 16$, and no dropout, introducing only ∼12M additional trainable parameters, which account for approximately 0.17% of the full backbone. In total, ∼142M parameters are updated during training, compared to ∼7B in full fine-tuning, resulting in a ∼98% reduction in the trainable parameters. Only these LoRA parameters, along with the image encoder, depth encoder, point cloud encoder, and the dense decoder are updated, while all remaining backbone weights are kept frozen.

Supervised fine-tuning (SFT) is performed on Scene-30K for 2 epochs with a batch size of 12, adopting the AdamW optimizer with a weight decay of 0.1 and a cosine annealing learning rate schedule that decays from $10^{-5}$ to $10^{-6}$. Following supervised fine-tuning (SFT), we further optimize the model via reinforcement learning using Group Relative Policy Optimization (GRPO). The RL stage is performed for 2 epochs with a batch size of 12, employing the Adam optimizer and a fixed learning rate of $10^{-6}$. To ensure stability, a KL divergence penalty with coefficient $\beta = 0.02$ is imposed between the current policy and the frozen SFT model.

Furthermore, we introduce a dynamic view selection strategy applied during both training and inference. Given a 3D scene with a pool of rendered multiview images, we extract visual features for each view using a pretrained SigLIP-2 encoder. For each view, we compute three complementary scores, which are aggregated using learnable weights to derive a final utility score. Following prior work (Luo et al., 2024), we select the top-6 views ranked by this score and feed them into the model alongside corresponding depth inputs. All experiments are conducted on $4 \times$ NVIDIA H20 GPUs.

### 3.3 MAIN RESULTS

**3D scene dense captioning.** It demands a model to localize and describe an object in a 3D scene. We compare SOTA methods on the widely used ScanRefer (Chen et al., 2020) and Nr3D (Achlioptas et al., 2020) benchmarks. The results in Table 2 show that our method consistently outperforms existing methods on both datasets.

**3D object captioning.** This task requires the model to describe a localized object in a 3D scene. We compare SOTA methods on Cap3D (Luo et al., 2023) benchmark. As shown in Table 4, "Allviews 28-views" indicates DiffuRank (Luo et al., 2024) trained with all 28 views, "Horizontal 6-views" with 6 horizontal views, "Bottom 6-views" with 6 bottom views. The results show that 3D-R1 achieves the highest scores across all evaluation criteria.

**3D question answering.** It requires a model to generate responses to the natural language queries questioning towards a 3D scene. We compare SOTA methods on the ScanQA (Azuma et al., 2022) validation set as well as two test benchmarks in Table 3. The results show that our method consistently outperforms existing methods on all evaluation sets.

**3D visual grounding.** It requires a model to accurately localize the object referenced by a natural language expression within a 3D scene. We benchmark state-of-the-art methods on the widely used Nr3D (Achlioptas et al., 2020) and ScanRefer (Chen et al., 2020) datasets as seen in Table 6. We can see that our method consistently outperforms existing methods on both datasets.

**3D reasoning.** It requires the model to infer spatial or functional relationships between objects based on contextual cues within a 3D scene. We evaluate on the SQA3D (Ma et al., 2023) benchmark and report standard metrics in Table 5. The results show that 3D-R1 achieves the highest scores across all metrics.

**3D dialogue.** This task involves generating interactive context-aware responses grounded in the 3D scene. We compare our method on the 3D-LLM (Hong et al., 2023) dataset, as shown in Table 5. 3D-R1 significantly outperforms previous models, achieving state-of-the-art results across all evaluation metrics.

**3D planning.** This task aims to generate sequential action plans based on instructions and 3D contextual understanding. We evaluate on the 3D-LLM (Hong et al., 2023) dataset. As reported in Table 5, 3D-R1 surpasses all baselines across all evaluation criteria.

## 4 CONCLUSION

In this work, we propose 3D-R1, a generalist 3D vision-language model designed to advance unified scene understanding. To address the shortcomings of existing 3D-VLMs in reasoning generalization, we introduce Scene-30K, a large-scale, high-quality Chain-of-Thought dataset that provides structured supervision for cold start initialization. Based on this foundation, we develop a reinforcement learning framework based on Group Relative Policy Optimization (GRPO), integrating perception-based, semantics-based, and format-based rewards to refine the model's cognitive alignment and spatial precision. In addition, we present a dynamic view selection strategy that learns to rank multiview images based on task relevance, spatial coverage, and cross-modal alignment. Extensive evaluations across seven representative 3D benchmarks demonstrate that 3D-R1 achieves significant improvements over prior methods. Our results highlight the promise of combining structured CoT supervision, reward-driven policy optimization, and adaptive perception strategies for generalizable 3D scene understanding.

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

APPENDIX

# A  RELATED WORK

**3D scene understanding.**  3D scene understanding targets the comprehension of the semantic meaning of objects and their surrounding environment through the analysis of point clouds. In this study, we focus on several integral scene understanding tasks: 3D Scene Dense Captioning (3D-DC), 3D Object Captioning, 3D Question Answering (3D-QA), 3D Dialogue, 3D Visual Grounding (3D-VG), 3D Reasoning, and 3D Planning. 3D-DC involves producing descriptive language based on a 3D environment, encompassing both individual objects and the entire scene. At the object level, models localize individual objects in a point cloud and describe each with natural language. Scan2Cap (Chen et al., 2021b) first introduced this task by detecting objects in RGB-D scans and generating context-aware captions for each. Subsequent work shifted from a two-stage "detect-then-describe" pipeline to an end-to-end transformer model. For example, Vote2Cap-DETR (Chen et al., 2023b) and its Vote2Cap-DETR++ (Chen et al., 2024b) variant use a DETR-based encoder–decoder to jointly detect and caption objects in one pass. At the scene level, models generate holistic descriptions of entire environments. The recent 3D-CoCa framework (Huang et al., 2025a) integrated contrastive vision language pretraining with caption generation to produce semantically coherent scene descriptions Huang et al. (2025b). Likewise, LLM-augmented methods, such as LSceneLLM (Zhi et al., 2024) incorporated global context and language priors and used an LLM's attention to focus on task-relevant areas and describe large cross-room scenes.

3D-QA extends the visual QA paradigm into 3D scenes, requiring spatial and cross-modal reasoning beyond 2D capabilities. The ScanQA (Azuma et al., 2022) benchmark introduced this task by pairing 3D indoor scans with questions. The follow-up work has increased the complexity, SQA3D (Ma et al., 2023), for example, situated an embodied agent in the scene and poses questions about the agent's surroundings, testing the model's ability to interpret the agent's viewpoint and reason about spatial relations in the 3D environment.

3D-VG focuses on locating referred objects in a 3D scene based on natural language expressions, requiring precise semantic and spatial alignment across modalities. Recent research advances have explored unified transformer-based architectures and LLM-enhanced grounding. 3DVG-Trans (Lichen et al., 2021) proposed a cross-modal transformer that fuses linguistic and point cloud level geometric features within a transformer-based framework. Building on the capabilities of large language models, GPT4Scene (Qi et al., 2025) explored the zero-shot grounding setting. It integrated GPT-4 with 3D feature encoders via a lightweight alignment module and prompted the LLM to resolve spatial references from language alone.

Reinforcement learning (RL) techniques have recently been introduced to further improve multi-modal 3D reasoning. (Chen et al., 2025) proposed to compile scene graphs with RL-enhanced MLLM, in a system called R1-SGG. They first train a multimodal LLM to output structured scene graphs from images and then refine it via RL with graph-centric rewards that promote high recall and semantic alignment of predicted objects and relationships. In a related vein, (Park et al., 2025) introduced DIP-R1, an RL-based framework that guides a multimodal LLM to perform fine-grained visual inspection in complex scenes. These investigations showcase the potential of RL to improve 3D scene understanding in conjunction with large vision language models.

**3D vision language models.**  Research on 3D vision–language models (3D-VLMs) has advanced rapidly, fueled by progress in large language models (LLMs). The early 3D-VLMs focused on understanding 3D object point clouds (Xu et al., 2024; Tang et al., 2024). PointLLM (Xu et al., 2024) introduced an initial 3D-VLM that couples a point cloud encoder with an LLM, enabling the model to interpret colored object point clouds and answer questions about the shape and attributes of an object. Another line of work, MiniGPT-3D (Tang et al., 2024) proposed an efficient strategy to align 3D data with language models utilizing 2D vision language priors.

More recently, researchers have shifted toward scene-level 3D-VLMs that can handle entire rooms or complex scenes with many objects. For example, LLaVA-3D (Zhu et al., 2024) augmented image patches with 3D position embeddings and performs joint 2D-3D instruction tuning, enabling the model to understand a whole scene and even output structured spatial information without relying on external detectors. A recent work, 3D-LLaVA (Deng et al., 2025) takes a complementary approach,

using a minimalist point-cloud-based pipeline with an integrated Omni Superpoint Transformer that acts as a visual encoder and multi-task decoder; this module selects salient 3D features, embeds interactive visual prompts, and can output grounded 3D segmentation masks, all within a single unified architecture.

# B ABLATION STUDY

**Reinforcement learning.** We conduct a comprehensive ablation to examine the effect of each reward function in our GRPO-based reinforcement learning. As presented in Table 7, reinforcement learning (RL) yields substantial improvements in both reasoning and grounding performance compared to the baseline of supervised fine-tuning (SFT). Although SFT provides strong initialization, it lacks structural regularity, spatial alignment, and semantic fidelity. The format reward enforces syntactic consistency in the output, the perception reward enhances spatial grounding through improved object localization, and the semantic reward improves alignment with the intended meaning. When combined, these reward signals lead to

Table 7: **Ablation of individual and combined rewards in GRPO-based RL.** Performance is evaluated on 3D-QA (ScanQA) and on 3D-DC (ScanRefer) tasks. And the first row corresponds to the supervised fine-tuning (SFT) baseline without any reinforcement learning.

| $R_{Format}$ | $R_p$ | $R_{similarity}$ | ScanQA | | ScanRefer | |
|---|---|---|---|---|---|---|
| | | | C↑ | R↑ | C@0.25↑ | R@0.25↑ |
| ✗ | ✗ | ✗ | 97.95 | 45.12 | 85.20 | 55.94 |
| ✓ | ✗ | ✗ | 101.35 | 46.65 | 88.00 | 57.52 |
| ✗ | ✓ | ✗ | 102.55 | 47.34 | 88.70 | 58.24 |
| ✗ | ✗ | ✓ | 102.45 | 47.50 | 88.50 | 58.33 |
| ✓ | ✓ | ✗ | 104.12 | 48.90 | 89.90 | 59.75 |
| ✓ | ✗ | ✓ | 104.75 | 49.03 | 90.20 | 59.84 |
| ✗ | ✓ | ✓ | 104.60 | 49.10 | 90.10 | 59.90 |
| ✓ | ✓ | ✓ | **106.45** | **51.23** | **91.85** | **62.23** |

a significant performance increase, increasing ScanQA CIDEr from 97.95 to 106.45 and ScanRefer C@0.25 from 85.20 to 91.85. This highlights the complementary contributions of each reward component in optimizing the model's 3D reasoning capabilities.

**Dynamic view selection.** To quantify the effect of dynamic view selection, we compare our learned strategy against three fixed-view baselines: (1) **All-views**, which uses all views of the scene; (2) **Horizontal 6-views**, comprising six front-facing views of the scene; and (3) **Bottom 6-views**, sampled from below the scene. In contrast, (4) **Ours (Learned 6-view selection)** adaptively selects the most informative six views based on learned utility scores. As shown in Table 8, our dynamic view selection strategy consistently outperforms fixed-view baselines. On the 3D object captioning

Table 8: **Effect of dynamic view selection.** Comparison of different view selection strategies for 3D object captioning (Cap3D) and 3D-VG (ScanRefer). Our learned selection of six optimal views achieves superior performance over fixed-view baselines.

| View Strategy | Cap3D | ScanRefer | |
|---|---|---|---|
| | CLIP R@1↑ | Acc@0.25↑ | Acc@0.5↑ |
| All-views | 29.19 | 61.25 | 51.73 |
| Horizontal 6-views | 30.18 | 60.53 | 50.26 |
| Bottom 6-views | 6.63 | 57.89 | 47.63 |
| **Learned 6-view selection (Ours)** | **32.23** | **65.85** | **59.24** |

task, it improves CLIP R@1 from 30.18 with fixed horizontal 6 views to 32.23, highlighting its ability to focus on more informative visual perspectives. Moreover, the performance gains observed on 3D visual grounding further demonstrate that adaptive view selection leads to more accurate object localization by providing contextually relevant observations.

We also study the effect of three dynamic view selection weights, which control the fusion of three scoring cues: text relevance ($w_t$), spatial coverage ($w_c$), and CLIP-based similarity ($w_{\text{clip}}$). Table 9 presents a grid search for various weight combinations. The results show that all three cues are complementary: using any single score alone yields suboptimal results, while balanced weighting ($w_t = 0.3$, $w_c = 0.5$, $w_{\text{clip}} = 0.5$) achieves the best performance across tasks.

Table 9: **Grid search on view weight configurations.** Performance is evaluated on 3D-QA (ScanQA) and on 3D-VG (ScanRefer) tasks.

| View weight | | | ScanQA | | ScanRefer | |
|---|---|---|---|---|---|---|
| $w_t$ | $w_c$ | $w_{\text{clip}}$ | C↑ | B-4↑ | Acc@0.25 | Acc@0.5 |
| 0.3 | 0.6 | 0.4 | 122.76 | 12.98 | 55.34 | 42.98 |
| 0.3 | 0.4 | 0.6 | 128.49 | 15.34 | 60.45 | 50.23 |
| 0.4 | 0.5 | 0.5 | 137.78 | 22.23 | 63.98 | 57.95 |
| 0.2 | 0.5 | 0.5 | 136.67 | 22.80 | 60.45 | 55.94 |
| **0.3** | **0.5** | **0.5** | **138.67** | **23.56** | **65.85** | **59.24** |

To further illustrate this, Figure 5 visualizes the performance landscape over different weight configurations. The plots reveal that moderate reliance on text grounding ($w_t \approx 0.3$–$0.4$) combined with

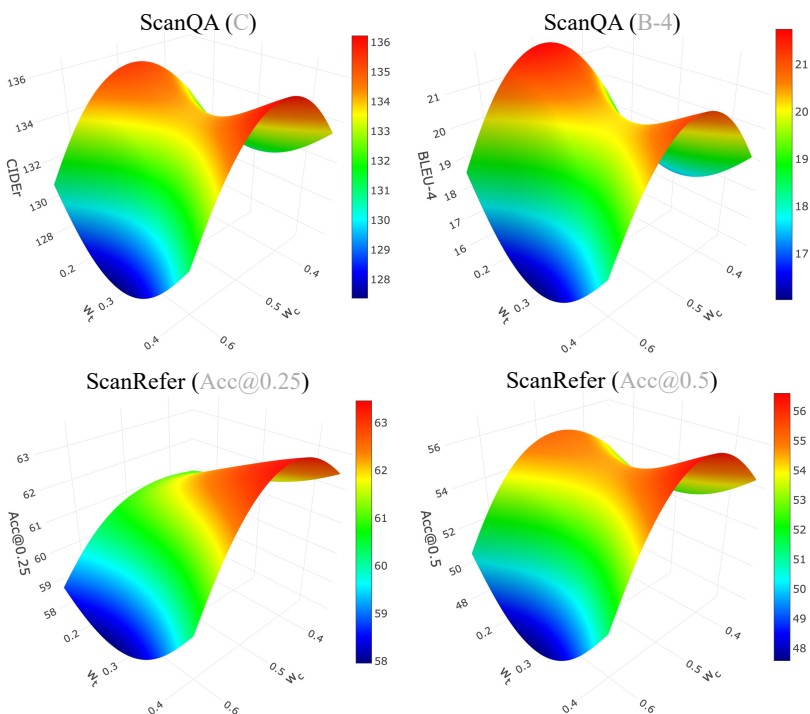

Figure 5: **Performance surfaces under different dynamic view selection weight configurations.** We analyze the influence of text relevance ($w_t$), spatial coverage ($w_c$), and CLIP-based similarity ($w_{\text{clip}}$) on model performance, with the constraint $w_c + w_{\text{clip}} = 1$. Results on 3D-QA (ScanQA) and 3D-VG (ScanRefer) reveal that optimal performance emerges when $w_t$ is within the range of 0.3 to 0.4, combined with balanced visual weights.

balanced visual cues leads to optimal performance, validating the effectiveness of learned weight fusion.

**Architecture and hyperparameters.** We conduct a step-by-step ablation to quantify the contribution of each modality encoder in our unified 3D architecture. As shown in Table 10, we start from a baseline model using only the text and image encoder, and progressively add the depth encoder and point cloud encoder. Each modality brings clear performance gains on both 3D reasoning (SQA3D) and 3D planning (3D-LLM) tasks. Adding the depth encoder improves performance on SQA3D, confirming that monocular geometric cues are helpful for grounding and planning. Further adding the point cloud encoder leads to larger gains, highlighting the importance of explicit 3D structure for complex reasoning. The full model (3D-R1) achieves the best performance across all metrics.

Table 10: **Incremental modality encoder ablation starting from Text & Image encoder.** Performance is evaluated on 3D reasoning (SQA3D) and 3D planning (3D-LLM) tasks. The first row is the baseline, and each subsequent row adds one encoder. The final row (3D-R1) includes all modalities.

| Setting | SQA3D | | 3D-LLM | |
|---|---|---|---|---|
| | C↑ | B-4↑ | C↑ | B-4↑ |
| Text & Image Encoder | 110.23 | 15.34 | 200.45 | 20.15 |
| + Depth Encoder | 115.23 | 18.34 | 205.45 | 21.15 |
| + Point Encoder | 120.12 | 20.13 | 215.34 | 22.34 |
| **3D-R1 (Ours)** | **138.67** | **23.56** | **230.50** | **25.45** |

Finally, we examine the impact of the LoRA rank $\delta$, which controls the internal dimensionality of the adapter layers. A higher rank allows for more expressive adaptation but increases the number of trainable parameters. As shown in Table 11, increasing $\delta$ from 4 to 12 results in significant performance gains across reasoning and grounding tasks, with ScanQA CIDEr improving from 94.57 to 106.45, and Nr3D accuracy rising from 63.12 to 68.80. However, the performance gains begin to saturate beyond $\delta = 12$, as further increasing the rank to 32 yields only marginal improvements at

Table 11: **Ablation of LoRA rank** $\delta$. Increasing rank improves performance up to a point, with diminishing returns beyond $\delta = 12$. Performance is evaluated on 3D-QA (ScanQA) and on 3D-VG (Nr3D) tasks.

| LoRA Rank $\delta$ | Params (M) | ScanQA | | | | Nr3D |
|---|---|---|---|---|---|---|
| | | C↑ | B-4↑ | M↑ | R↑ | Acc@0.25 |
| 4 | 82 | 94.57 | 13.34 | 17.12 | 47.23 | 63.12 |
| 8 | 112 | 101.69 | 15.34 | 20.12 | 49.23 | 65.43 |
| **12 (Ours)** | 142 | **106.45** | **17.80** | **22.13** | **51.23** | **68.80** |
| 16 | 175 | 106.79 | 17.45 | 22.23 | 51.33 | 68.82 |
| 32 | 250 | 107.01 | 17.90 | 22.50 | 51.45 | 68.90 |

the cost of higher parameter overhead. These results suggest that $\delta = 12$ offers the best trade-off between performance and efficiency.

## C  IMPLEMENTATIONS DETAILS

**Data synthesis.** First, a Scene-30K dataset is synthesized using Gemini-Pro Team et al. (2025), producing 35,248 raw CoT reasoning examples. To ensure that only high-quality chains of thought (CoT) are retained, we design a rule-based filtering that reduces the corpus to 30,012 examples. Some examples are visualized in Figure 6-10.

Specifically, the rule-based filtering process is as follows: We first verify that each example follows the required output format: `<think>reasoning</think><answer>final answer</answer>`. The `<think>` segment must contain least 30 words, and the `<answer>` segment at least 20 words, to filter out overly brief reasoning and answers. Subsequently, we assess whether the `<think></think>` segment exhibits genuine multi-step reasoning, as opposed to a single-step deduction. To ensure this, we mandate the presence of at least three explicit reasoning steps, identified through lexical cues such as "Step n", "First", "Next" or "Last". Moreover, the final step must explicitly reference the target entity posed in the question (*e.g.*, "Conclusion: ..."), as exemplified in Figure 6–10. Finally, we assess the logical consistency between the reasoning and the answer. Specifically, we prompt Gemini 2.5 Pro Team et al. (2025) with the pair {*think*, question}, where *think* refers to the reasoning content enclosed within the `<think></think>` tags. The model is asked to independently generate an answer $\hat{a}$. A sample is retained only if the normalized Levenshtein similarity between $\hat{a}$ and the content within the `<answer></answer>` tags, denoted as $a$, is at least 0.8. The similarity score is defined as:

$$\text{Similarity}(\hat{a}, a) = 1 - \frac{D_{\text{lev}}(\hat{a}, a)}{\max(|\hat{a}|, |a|)}, \tag{8}$$

where $D_{\text{lev}}(\hat{a}, a)$ denotes the Levenshtein distance, and $|\cdot|$ represents the character length of the string. If the score falls below 0.8, the sample is discarded, even if it satisfies the format and step-count criteria.

The complete filtering procedure is summarized in Algorithm 1. After applying all filtering criteria, Scene-30K dataset is constituted and serves as the cold-start initialization for 3D-R1.

## D  VISUALIZATION

To qualitatively assess the capabilities of 3D-R1 in various 3D scene understanding tasks, we provide visualizations in Figures 11-17. These examples highlight the reasoning ability of the model, spatial comprehension, and multimodal alignment.

## E  LIMITATION AND FUTURE WORK

While 3D-R1 achieves strong reasoning performance and generalizability across diverse 3D scene understanding tasks, several limitations remain. First, although the Scene-30K dataset provides

**Algorithm 1** Rule-based Filtering for Scene-30K

**Require:** Raw CoT examples $\mathcal{D}_{\text{raw}} = \{(q_i, t_i, a_i)\}_{i=1}^{N}$
**Ensure:** Filtered CoT dataset $\mathcal{D}_{\text{final}}$
1: $\mathcal{D}_{\text{final}} \leftarrow \emptyset$
2: **for all** $(q, t, a)$ in $\mathcal{D}_{\text{raw}}$ **do**
3:     **if** format is invalid **then**
4:         **continue**
5:     **end if**
6:     **if** word count of $t < 30$ or word count of $a < 20$ **then**
7:         **continue**
8:     **end if**
9:     **if** number of reasoning steps in $t < 3$ **then**
10:         **continue**
11:     **end if**
12:     **if** final step in $t$ does not mention target entity **then**
13:         **continue**
14:     **end if**
15:     Prompt Gemini 2.5 Pro with $(t, q)$ to generate predicted answer $\hat{a}$
16:     Compute Levenshtein similarity score: $s = 1 - \frac{D_{\text{lev}}(\hat{a}, a)}{\max(|\hat{a}|, |a|)}$
17:     **if** $s < 0.8$ **then**
18:         **continue**
19:     **end if**
20:     Add $(q, t, a)$ to $\mathcal{D}_{\text{final}}$
21: **end for**
22: **return** $\mathcal{D}_{\text{final}}$

high-quality Chain-of-Thought (CoT) supervision, it is primarily synthetic and may not fully capture the richness and ambiguity of real-world human reasoning. Second, the current GRPO-based RLHF optimization operates at the response level and lacks temporally grounded feedback, limiting the model's ability to reason and act over long horizons in embodied settings.

In future work, we plan to extend 3D-R1 in two key directions. First, we will explore embodied AI applications that integrate path planning and action prediction with multimodal reasoning. Second, we aim to develop a world model atop 3D-R1, enabling agents to simulate and predict future scene dynamics for more robust decision-making.

# F   LLM USE DECLARATION

Large Language Models (ChatGPT) were used exclusively to improve the clarity and fluency of English writing. They were not involved in research ideation, experimental design, data analysis, or interpretation. The authors take full responsibility for all content.

## Prompt

You are an AI visual assistant in a 3D scene. Each scene contains a piece of description as follows.

Scene description of the scene: In this apartment scene, there is a floor, sink, mirror, desk, clock, scale, kitchen cabinets, cabinets, tables, toaster, stools, bed, trash cans, dish rack, curtains, tissue box, toilet, bicycle, shelf, and a guitar case. The sink is in front of the guitar case, while the cabinet is to the left of the guitar case. The stool is in front of the scale and the toilet. The bicycle is adjacent to the cabinet, and the shelf is in front of the sink. The floor supports the stool and the sink. The scene suggests a functional and organized apartment with various objects for daily activities.

**Following the scene description, think step by step and answer the following question directly with following output format**.

**Output format:** <think>... reasoning...</think> <answer>... final answer...</answer>.

**Question:** What is in the right corner of room by curtains?

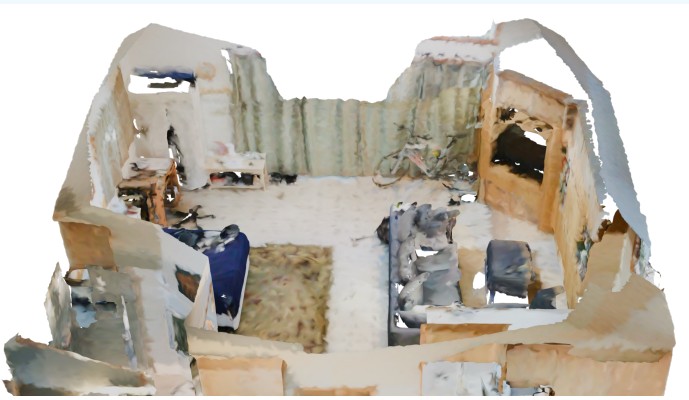

## Response

<think>
**Step 1:** Locate the spatial reference
**"*right corner by curtains*"** The question specifies the right corner of the room and objects near the curtains.
This sets our spatial query target.

**Step 2:** Identify all mentions of curtains and associated objects
A green curtain is described as behind a bicycle.
A tan curtain is above a desk near the bed.
Of these, the green curtain is more likely to be in a corner, since the desk bed setup implies a central or side-wall arrangement.

**Step 3:** Explore the green curtain region
Near the green curtain: a bicycle and a cabinet. The cabinet is described as housing a TV, and placed opposite the kitchen and sofa.
This implies a corner-located entertainment area.

**Step 4:** Evaluate dominance of objects in that corner
The bike is adjacent, but the TV cabinet is structurally larger, described as a focal point, and clearly fixed in place.
Curtains likely frame the cabinet setup, reinforcing its positional centrality in that corner.

**Conclusion:** The **TV cabinet** is the primary object in the right corner near the **green curtains**.
</think>
<answer>
The object in the right corner of the room by the curtains is the **TV cabinet**, which contains the TV.
</answer>

Figure 6: **Scene-30K CoT data example 1.**

## Prompt

You are an AI visual assistant in a 3D scene. Each scene contains a piece of description as follows.

Scene description of the scene: In the kitchen, there is a sink, an armchair, a kitchen counter, two windows, two trash cans, two refrigerators, a floor, and ten chairs. The chairs are positioned in front of each other and behind the trash cans. The trash cans are in front of the chairs, and the chairs are to the right of the armchair and another chair. The chairs are supported by the floor. The kitchen appears to be a functional space with various objects for cooking, sitting, and disposing of waste.

**Following the scene description, think step by step and answer the following question directly with following output format.**

**Output format:** <think>... reasoning...</think> <answer>... final answer...</answer>.

**Question:** What color is the refrigerator?

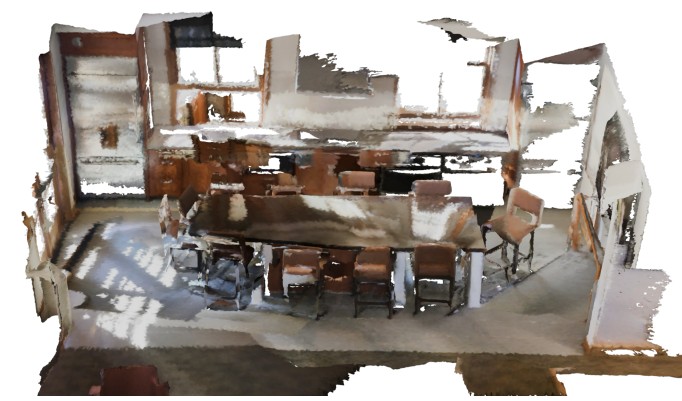

## Response

<think>
**Step 1:** Identify the query and object
The question is: **"*What color is the refrigerator?*"** This requires identifying all refrigerators in the scene and resolving their associated color attributes.
This sets our spatial query target.

**Step 2:** Collect all references to refrigerators and colors
A stainless steel refrigerator is described with a bottom freezer.
A white refrigerator is also mentioned.
This suggests that the scene may contain more than one refrigerator or that there is ambiguity in how a single unit is described.

**Step 3:** Determine the number of refrigerators
One caption explicitly states "two refrigerators", which strongly implies the existence of two separate appliances.
Thus, the color differences likely correspond to two distinct refrigerators rather than conflicting descriptions of one.

**Step 4:** Evaluate user intent and precision of the query
The question is phrased as "the refrigerator", suggesting singular intent.
However, since the scene contains two refrigerators, both should be acknowledged unless additional disambiguation is given.

**Conclusion:** The kitchen contains **two refrigerators**: one is **stainless steel**, and the other is **white**.
</think>
<answer>
The **refrigerators** in the kitchen are of different colors. One is **stainless steel**, and the other is **white**.
</answer>

Figure 7: **Scene-30K CoT data example 2.**

## Prompt

You are an AI visual assistant in a 3D scene. Each scene contains a piece of description as follows.

Scene description of the scene: In this classroom scene, there is a single floor, three windowsills, 22 chairs, two windows, and seven tables. The chairs are adjacent to the tables, with some chairs behind the tables and others to the left. The tables are positioned in front of some chairs and behind others. The tables are also stacked, with one table behind another. The chairs are resting on the floor, which supports both the chairs and the tables. Inside one of the tables, there is a chair. The classroom appears to be well-equipped with furniture for students to sit and work comfortably.

**Following the scene description, think step by step and answer the following question directly with following output format.**
**Output format:** <think>... reasoning...</think> <answer>... final answer...</answer>.

**Question:** Please describe the table.

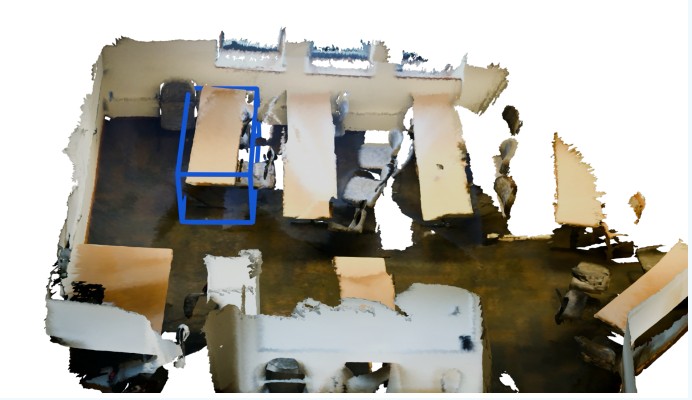

## Response

<think>
**Step 1:** Localize the queried table
The question refers to **"*the table*"**, and the blue 3D bounding box highlights a specific rectangular table placed in the classroom scene.
Given the presence of seven tables, this selection implies that the referred table is either spatially unique, visually distinct, or structurally notable.

**Step 2:** Analyze spatial context and adjacency
This table is positioned **near the wall and windows**, suggesting it is in a front or peripheral position relative to the rest of the layout.

**Step 3:** Interpret table arrangement and visibility
Its orientation is consistent with the row-style classroom layout. The surface color is darker than adjacent tables, possibly due to lighting, material, or rendering artifacts, which may imply a distinct role or usage.

**Step 4:** Evaluate its role and prominence
Being placed near the windows and slightly apart from the central clutter of chairs suggests this table may be used by an instructor or for special functions.

**Conclusion:** The table highlighted in blue box is a **rectangular classroom table**, located near the wall and windows. It likely serves an active role in the classroom, possibly for instruction or individual work.
</think>
<answer>
The table is a **rectangular classroom table**, located near the wall and windows.
</answer>

Figure 8: **Scene-30K CoT data example 3.**

## Prompt

You are an AI visual assistant in a 3D scene. Each scene contains a piece of description as follows.

Scene description of the scene: In this office scene, there are two windows, one pair of headphones, two file cabinets, one telephone, and five monitors. The monitors are positioned adjacent to each other and beside another monitor. Interestingly, the headphones are found inside one of the monitors. The office appears to be well-equipped with technology, suggesting a modern and efficient workspace. The presence of file cabinets implies a need for organization and storage. Overall, this office scene exudes a professional and productive atmosphere.

**Following the scene description, think step by step and answer the following question directly with following output format.**
**Output format:** <think>... reasoning...</think> <answer>... final answer...</answer>.

**Question:** What color does the sofa look?

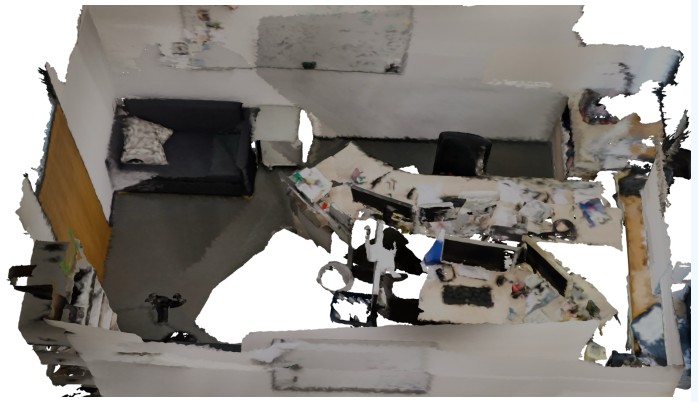

## Response

<think>
**Step 1:** Localize the queried table
The question asks: **"What color does the sofa look?"** This implies that we need to locate all references to the sofa in the scene and determine the most consistently described color.

**Step 2:** Extract sofa-related descriptions from captions
The sofa is referred to as a "*dark colored two-seater futon.*". Another line mentions a "*black sofa.*" These two phrases strongly suggest that the sofa is likely dark, with black being a probable color.

**Step 3:** Examine object-level annotations
Looking into the object annotations
One mentions a "*dark colored two-seater futon located by the door,*" Another describes a "*black sofa located to the wall behind the fan.*"
These consistently describe the sofa using the terms black, which suggests that the sofa appears consistently dark in the scene.

**Step 4:** Check for contradictory evidence
To ensure reliability, we examine whether any alternative colors are associated with the sofa. No such references are found, which strengthens the hypothesis that black is the intended color.

**Conclusion:** The **sofa** is most likely **black**.
</think>
<answer>
**The sofa** in the scene is **black**.
</answer>

Figure 9: **Scene-30K CoT data example 4.**

# Prompt

You are an AI visual assistant in a 3D scene. Each scene contains a piece of description as follows.

Scene description of the scene: In the Conference Room, there are two cabinets and twenty-five chairs. The cabinets are positioned to the left and behind the chairs, while the chairs are to the right of the cabinets. Additionally, one chair is in front of the cabinet, and another chair is to the left of it. The room appears to be a functional space for meetings or discussions, with the cabinets potentially used for storage and the chairs for seating. The arrangement suggests a professional and organized atmosphere.

**Following the scene description, think step by step and answer the following question directly with following output format**.
**Output format:** <think>... reasoning...</think> <answer>... final answer...</answer>.

**Question:** What is the object on the wall north of the yellow table?

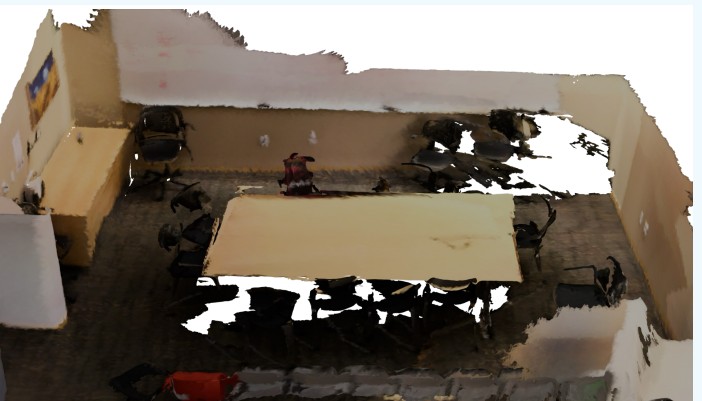

# Response

<think>
**Step 1:** Identify the query and spatial reference
The question asks: **"What is the object on the wall north of the yellow table?"** This implies a need to first determine the location of the yellow table and then identify the object directly north of it.

**Step 2:** Understand the scene layout
According to the scene description, the setting is a conference room, which contains: a yellow table, chairs, cabinets, and a whiteboard. The mention of a **whiteboard** in the scene immediately suggests it could be a candidate for a wall-mounted item in a directional context.

**Step 3:** Cross-reference with caption information
Scene captions state that *"the whiteboard is located next to the large table"*, and in one instance, it is said to be *"on the wall north of the yellow table."* This direct mention supports the hypothesis that the whiteboard lies on the north wall, aligning with the spatial query.

**Step 4:** Check for competing candidates
There are mentions of *cabinets* and *pictures*, but their described positions place them near corners or above other objects, not clearly on the north wall relative to the yellow table. This strongly implies that the whiteboard is the only clearly described object positioned on the wall north of the yellow table.

**Conclusion:** The object on the wall north of **the yellow table** is most likely **a whiteboard**.
</think>
<answer>
The object **on the wall north of the yellow table** is **a whiteboard**.
</answer>

Figure 10: **Scene-30K CoT data example 5.**

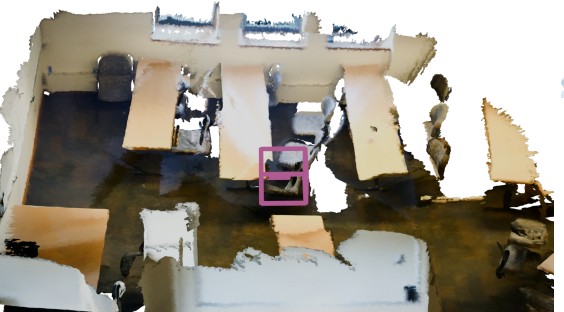

Please locate the TV in the scene and describe it.

The is a wall-mounted flat-screen display, landscape oriented with a dark glossy panel and slim bezels, fixed on the right wall above a narrow console near the corner and the faint reflections suggest the screen is off or at very low brightness, suitable for presentations or casual viewing.

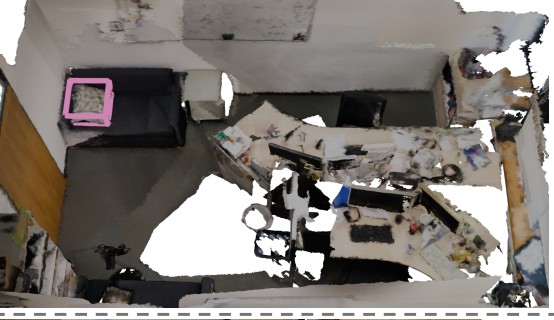

Please locate the chair in scene and describe it.

The object is a small chair with a light-colored cushioned seat and four wooden legs. It appears to have a very low or no visible backrest and is placed centrally between two seating areas. The chair is compact in size, making it suitable for occasional seating.

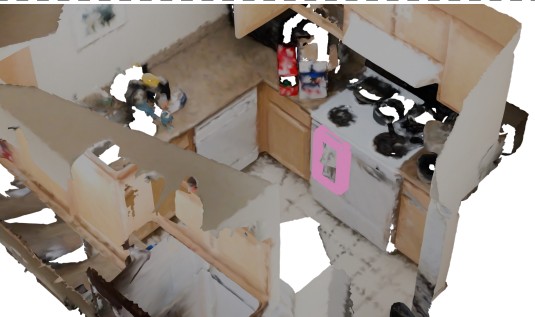

Please locate the pillow on the couch and describe it.

The object is a decorative pillow placed on the sofa. The pillow has a textured or patterned surface, possibly with a mix of light and dark tones, and it adds a soft contrast to the solid upholstery of the sofa. Positioned in the corner, it likely serves comfort purpose in the office lounge area.

Please locate the towel in the scene and describe it.

The object is a kitchen towel hanging on the handle of an oven door. It appears to be rectangular and made of fabric, likely used for drying hands or wiping surfaces. Its placement suggests it is positioned for convenient access while cooking at the stovetop or using the oven.

Figure 11: **Qualitative results for 3D scene dense captioning (3D-DC).**

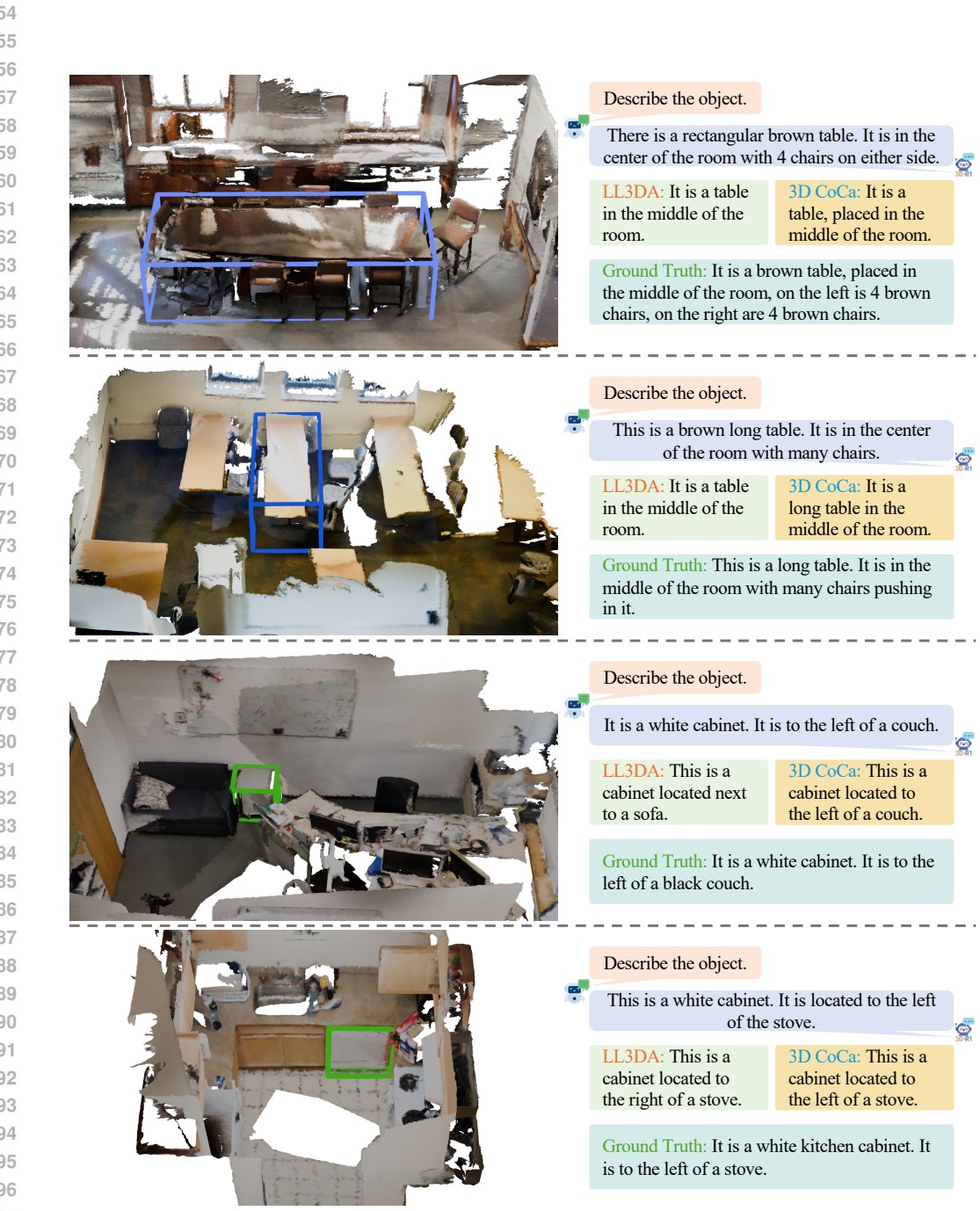

Figure 12: **Qualitative results for 3D object captioning.**

This is a black table lamp on the wall at the end of the bed.

Wooden nightstand with a white lamp, placed to the left of the bed.

Brown armchair on the rug near the back wall desk.

Figure 13: **Qualitative results for 3D visual grounding (3D-VG).**

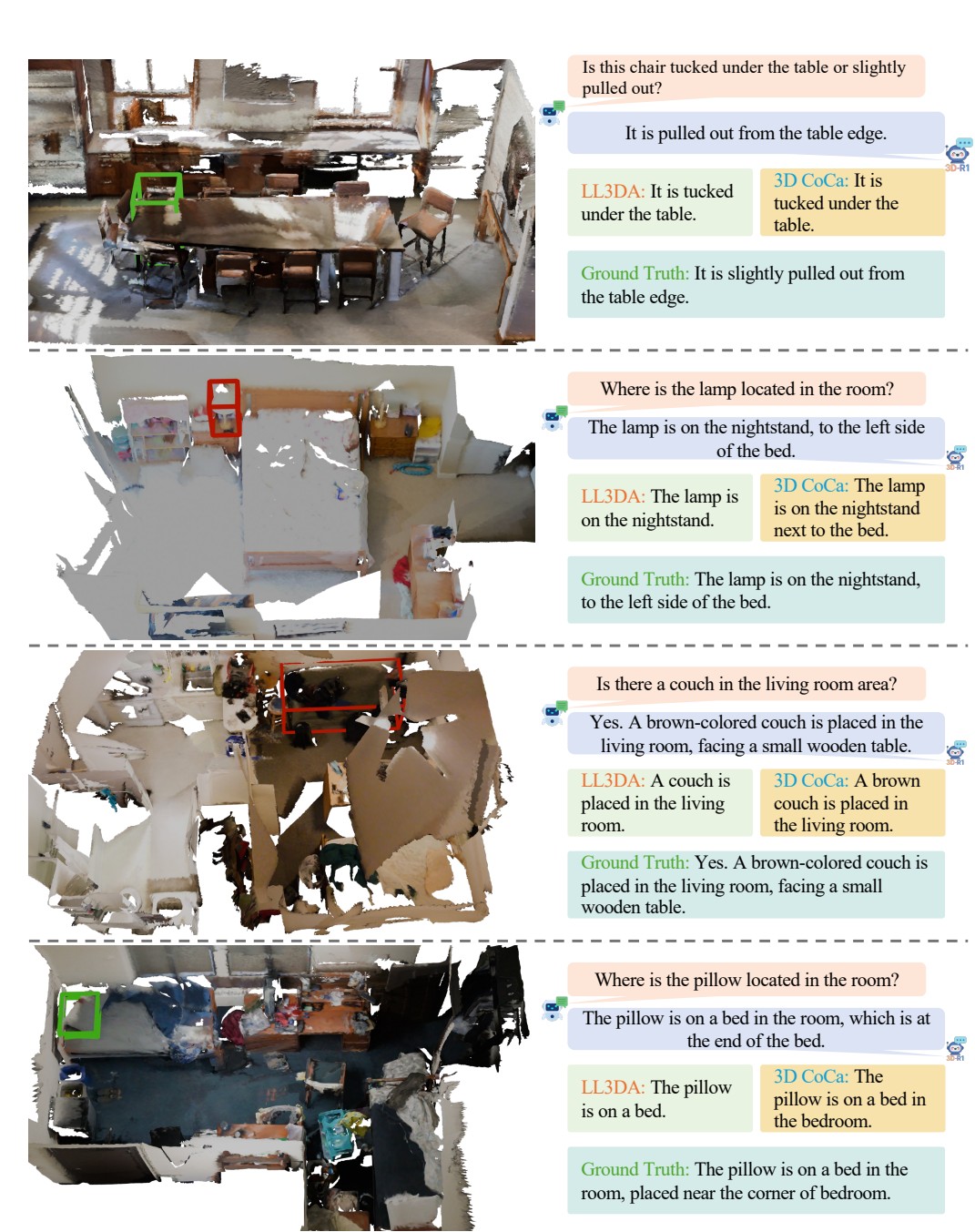

Figure 14: **Qualitative results for 3D question answering (3D-QA).**

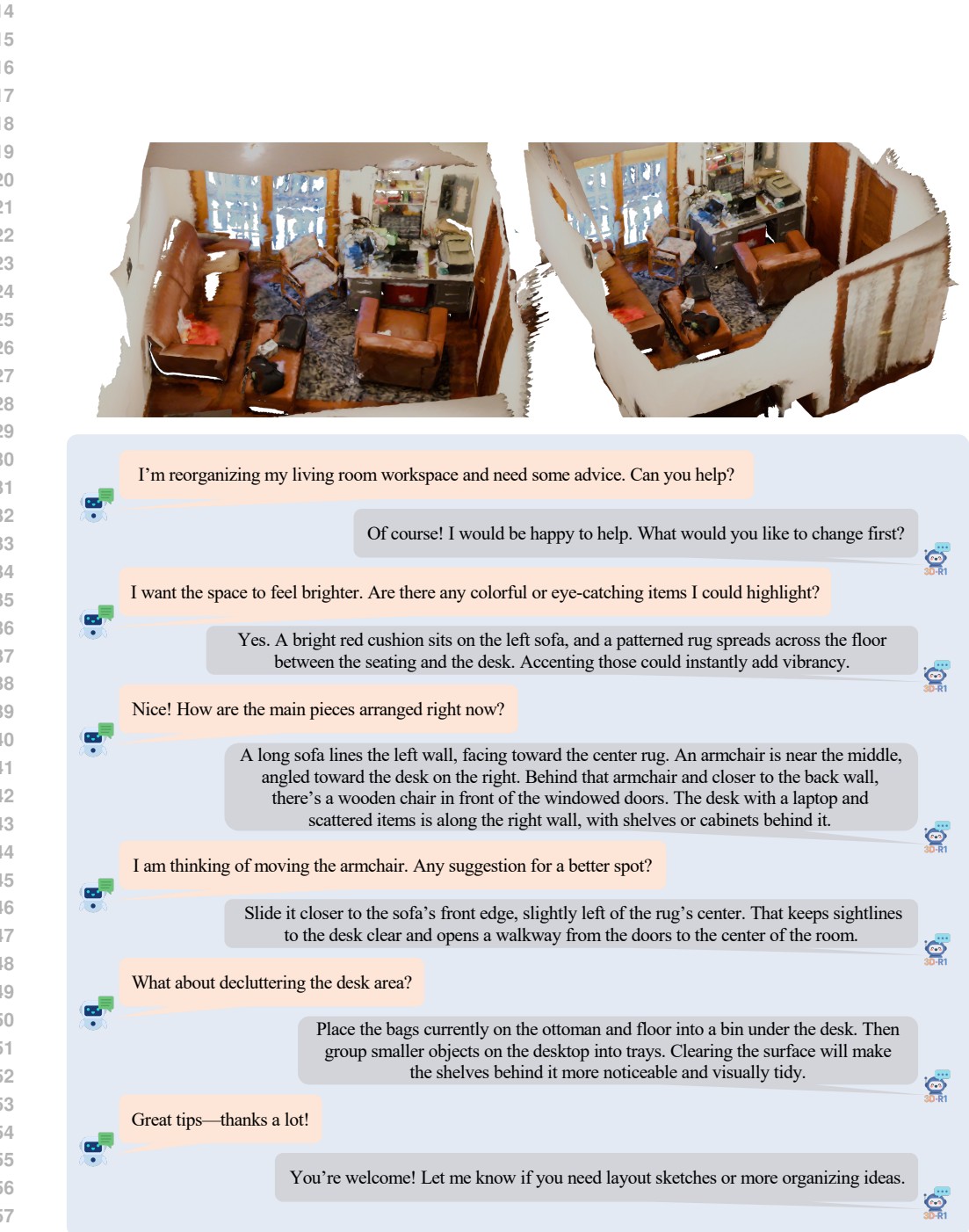

Figure 15: **Qualitative results for 3D dialogue.**

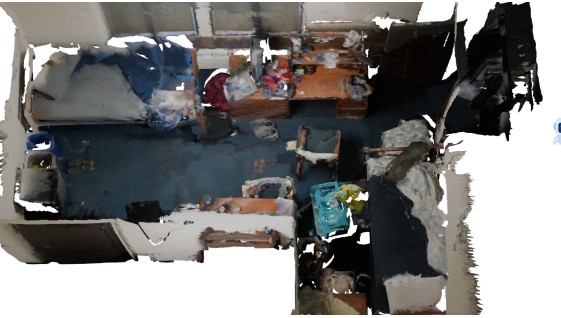

Standing in the doorway and facing the curved, clutter-covered workstation in the center of the room. Can I walk straight to the black office chair I see ahead of me without hitting anything?

No. Because the curved desk jutting out from the workstation, together with piles of papers and boxes on the floor, blocks a straight path—you'd collide with the desk edge before reaching the chair.

Standing just inside the doorway and facing the long workbench that runs beneath the window. Can I walk straight to the black floor-lamp by the window without hitting anything?

No. Because the white desk-peninsula with its drawer unit juts out between you and the lamp, blocking a direct path—you'd bump into the desk's corner before you reached the lamp.

Standing just inside the doorway at the foot of the bed and facing the long desk that runs beneath the window. Can I walk straight to that desk without tripping over anything?

No. Because the blue trash-bin and a scattered pile of clothes sit directly between you and the desk, a straight path is blocked—you'd bump the bin's rim and have to sidestep the clothes before you could reach the desk.

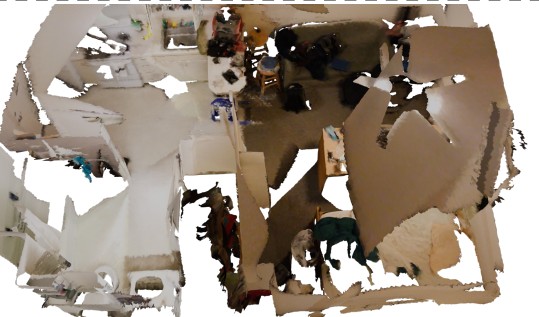

Sitting on the gray sofa and looking toward the kitchen, can I see the stovetop well enough to watch a boiling pot?

No. Because the kitchen peninsula and the short dividing wall form a high occluding edge between the living area and the back counter—your line of sight hits that column before the burners.

Figure 16: **Qualitative results for 3D reasoning.**

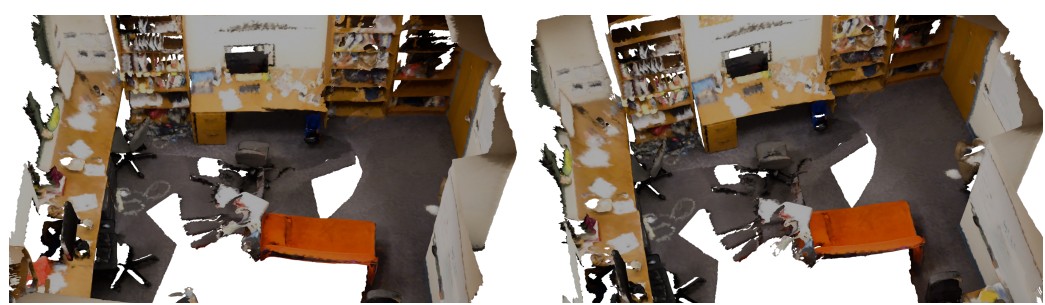

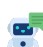 Please tidy and reorganize the workspace, keeping a clear central walkway.

1. Collect loose papers and gadgets on the long left-wall desk; slide every rolling chair fully under it.
2. Shift the gray swivel chair toward the left desk; tuck the small blue stool under the rear shelf to open a 1 m corridor.
3. Move the orange sofa back so its backrest is parallel to the bottom wall; clear the floor in front and bin scattered items.
4. Sort the back-wall shelves: heavy boxes low, frequently used items at chest height, rarely used on top; label storage bins.
5. Bundle and route cables behind the monitor on the rear desk; place chargers in one tray at the rear-right corner.
6. Vacuum starting from the right entrance, along the new corridor, around the sofa, and under both desks.
7. Final check: no chair legs or boxes protruding, drawers shut, whiteboard area unobstructed.

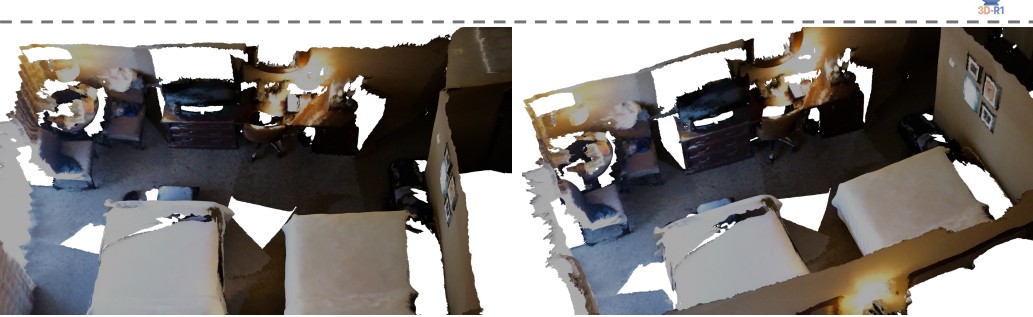

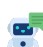 Reorganize the bedroom with a clean, direct corridor to the back-wall desk.

1. Bag trash, put laundry in a bin, and empty desk surface except monitor and one lamp.
2. Shift the right bed toward the bottom wall; pull the left bed toward the window wall.
3. Push the desk chair fully under the desk; slide the dresser flush to the right wall; set one nightstand between the beds; move the armchair and round table tight to the left wall as a reading nook.
4. Tuck the suitcase along the rear-right wall; bundle cables behind the monitor; chargers in the top right drawer.
5. Vacuum the aisle and bed perimeters; final check that nothing protrudes into the corridor and both lamps are reachable from bed and desk.

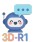

Figure 17: **Qualitative results for 3D planning.**

