# OpenReview forum: "3D-R1: Enhancing Reasoning in 3D VLMs for Unified Scene Understanding"
_ICLR.cc/2026/Conference — ICLR 2026 Conference Desk Rejected Submission_

### Official Review · Reviewer_NWsq · 2025-10-26

**Soundness:** 3
**Presentation:** 2
**Contribution:** 3
**Rating:** 6
**Confidence:** 4

**Summary:**

This paper presents 3D-R1, a 3D vision-language foundation model aimed at enhancing reasoning and generalization in unified 3D scene understanding. The model combines cold-start supervised fine-tuning on a newly built Scene-30K CoT dataset with reinforcement learning using Group Relative Policy Optimization (GRPO). Extensive experiments on multiple benchmarks (ScanRefer, Nr3D, Cap3D, ScanQA, 3D-LLM, and SQA3D) show consistent performance gains, with 3D-R1 outperforming prior 3D VLMs by roughly 10% on average across tasks.

**Strengths:**

1. The integration of reinforcement learning (GRPO) into 3D vision-language training is essential. The use of multi-reward signals to align reasoning, perception, and semantic accuracy is a clear conceptual advancement.
2. The Scene-30K dataset, generated with Gemini 2.5 Pro and structured CoT reasoning, is a valuable resource for promoting step-by-step spatial reasoning in 3D.
3. The experiments are comprehensive and detailed across diverse tasks.

**Weaknesses:**

1. The discussion of related work is insufficient. Several recent and more advanced 3D multimodal LLMs—such as **Inst3D-LMM (CVPR 2025)** and **Video-3D LLM (CVPR 2025)**—are not discussed or compared. A more comprehensive review and comparison would strengthen the paper.
2. The writing quality, as well as the presentation and layout of figures and tables, still have considerable room for improvement.
3. The paper lacks ablation studies to validate the generalization ability and effectiveness of the proposed 3D-R1.

**Questions:**

I will update my assessment after considering feedback and technical insights from other reviewers.

---

> ### Author Response · Authors · 2025-11-16
> **Response to Reviewer NWsq (1/2)**
>
> We thank Reviewer NWsq for the careful reading, for recognizing the importance of integrating GRPO-based reinforcement learning and the Scene-30K CoT dataset, and for noting the breadth of our experiments. We address the raised concerns below. In this first comment, we respond to the reviewer’s points on related work and writing or presentation; we discuss ablation studies and generalization in a second comment.
>
> ---
>
> ### 1. Related work and comparison with recent 3D multimodal LLMs
>
> **Reviewer’s concern.** "The discussion of related work is insufficient. Several recent and more advanced 3D multimodal LLMs—such as **Inst3D-LMM (CVPR 2025)** and **Video-3D LLM (CVPR 2025)**—are not discussed or compared."
>
> **Our response.** We appreciate this suggestion and agree that a more explicit positioning w.r.t. recent 3D LMMs will strengthen the paper.
>
> - **Video-3D LLM.** Video-3D LLM$^{[1]}$ treats 3D scenes as dynamic videos and augments video representations with 3D positional encoding for 3D scene understanding. In our submission, Video-3D LLM is already included as a strong baseline in our 3D visual grounding table (ScanRefer), where 3D-R1 achieves higher scores across the reported metrics (see the table that includes Video-3D LLM and GPT4Scene). **In the revision**, we will (i) explicitly mention this comparison in the main text when discussing those results, (ii) extend our comparisons by adding Video-3D LLM to the 3D scene dense captioning (3D-DC) and 3D-QA tables using the released results or our re-evaluation based on the official implementation, and (iii) add a short paragraph in the related-work section to clarify that Video-3D LLM focuses on video-based 3D representations, whereas 3D-R1 focuses on CoT-based reasoning, RLHF with multi-reward GRPO, and dynamic view selection over 3D scenes.
>
> - **Inst3D-LMM.** Inst3D-LMM$^{[2]}$ proposes an instance-aware 3D LMM that fuses multi-view 2D open-vocabulary priors with 3D geometry, and is particularly tailored to instance-level 3D-language tasks via multi-modal instruction tuning. This work is concurrent to ours and complementary in scope: while Inst3D-LMM emphasizes fine-grained instance tokens and multi-view fusion, 3D-R1 focuses on (i) a unified 3D generalist model covering dense captioning, QA, dialogue, planning, reasoning, grounding, and object captioning, and (ii) a CoT-driven training paradigm that combines a dedicated Scene-30K CoT dataset, GRPO-based RLHF with multi-reward signals, and dynamic 3D view selection.
>
> - **Revisions.** In the revised manuscript, we will (1) add a dedicated paragraph in the "3D vision-language models" part of the related-work section summarizing Inst3D-LMM and Video-3D LLM and contrasting them with 3D-R1; and (2) clarify in the experiments section that Video-3D LLM is already used as a baseline and that 3D-R1 remains competitive or superior on shared benchmarks. And, we will also add results of Inst3D-LMM on the benchmarks it supports (e.g., ScanRefer / ScanQA).
>
> ---
>
> ### 2. Writing quality and presentation/layout of figures and tables
>
> **Reviewer’s concern.** "The writing quality, as well as the presentation and layout of figures and tables, still have considerable room for improvement."
>
> **Our response.** We appreciate this feedback and will revise the paper for clarity and presentation:
>
> - **Improved exposition and structure.** We will streamline the description of the Scene-30K data engine (Sec. 2.2) and the GRPO-based RL pipeline (Sec. 2.4) to reduce repetition (e.g., avoid repeatedly re-introducing the \<think>/\<answer> format).
>
> - **Figures and captions.** We will improve the readability of Figures 2–5 by enlarging fonts, simplifying sub-figure labels, and making the captions more descriptive. For example, for Figure 2 we will clarify the roles of sub-figures (architecture, question-type distribution, multi-task performance, and generalizability) and explicitly refer to them in the text when we discuss cross-task generalization. We will also ensure the figure and table styles (color schemes, fonts, and spacing) are consistent across the paper.
>
> - **Proofreading.** We will perform a thorough proofreading pass to fix small grammatical issues (e.g., article usage, tense consistency) and improve sentence flow in the introduction, method, and experiment sections.
>
> We believe these changes will make the paper clearer and easier to follow. We address the reviewer’s concern about ablation studies and generalization in a follow-up comment (2/2).
>
> [1] Duo Zheng, Shijia Huang, and Liwei Wang. Video-3D LLM: Learning position-aware video representation for 3D scene understanding. In CVPR, 2025.
>
> [2] Hanxun Yu, Wentong Li, Song Wang, Junbo Chen, and Jianke Zhu. Inst3d-lmm: Instance-aware 3d scene understanding with multi-modal instruction tuning. In CVPR, 2025.

---

> ### Author Response · Authors · 2025-11-16
> **Response to Reviewer NWsq (2/2)**
>
> This comment is a continuation of our response to Reviewer NWsq. Here we address the concern about ablation studies and generalization.
>
> ---
>
> ### 3. Ablation studies and validation of effectiveness/generalization
>
> **Reviewer’s concern.** "The paper lacks ablation studies to validate the generalization ability and effectiveness of the proposed 3D-R1."
>
> **Our response.** Thank you for pointing this out. Our current submission does contain several ablations in **Appendix B**, but we agree that highlighting them more prominently in the main text will better support the claims.
>
> - **Existing ablations (currently in Appendix B).**
>   - *Reward design in GRPO.* Table 7 ablates the three GRPO rewards (format, perception, semantic-similarity) and shows that each component contributes and that combining all three gives the largest gains on ScanQA and ScanRefer.
>   - *Dynamic view selection.* Table 8 compares our learned dynamic view selection (six learned views) against fixed-view baselines and shows clear improvements on Cap3D and ScanRefer.
>   - *Architecture and modality encoders.* Table 10 performs an incremental ablation over the text+image, +depth, and +point-cloud encoders, demonstrating that each modality contributes to SQA3D and 3D-LLM performance, with the full 3D-R1 model achieving the best results.
>   - *LoRA rank.* We also study the impact of the LoRA rank $\delta$ and show that $\delta$ = 12 provides the best trade-off between performance and parameter efficiency.
>
> - **Revisions to better emphasize generalization and effectiveness.**
>   - We will move the most important ablation tables (reward ablation, dynamic view selection, and modality encoder ablation) from **Appendix B** into the main "Experiments" section under a dedicated “Ablation Study” subsection, and explicitly discuss how they support the effectiveness of 3D-R1.
>   - We will more clearly connect these ablations to the generalization results summarized in **Figure 2(c,d)**: Figure 2(c) shows that a single 3D-R1 model achieves strong performance across seven heterogeneous 3D tasks compared to prior methods, while Figure 2(d) further illustrates that our SFT+GRPO training paradigm consistently improves performance across these tasks, thereby validating the generalization ability and effectiveness of our design.
>
> We hope that surfacing these ablations in the main text and clarifying their connection to multi-task and cross-dataset performance will address the reviewer’s concern about effectiveness and generalization.
>
> ---
>
> We thank the reviewer again for the constructive comments. We believe that the outlined revisions—expanded related-work and comparisons, improved writing and figure or table presentation, and moving ablations into the main text—will significantly strengthen the paper.

---

> ### Author Response · Authors · 2025-11-22
> **Welcoming further feedback and discussion**
>
> Dear Reviewer,
>
> Thank you again for your valuable feedback and support. If we have addressed your concerns, we kindly ask you to consider raising your score. We are of course happy to respond to any further questions or suggestions you may have.
>
> Best regards,
>
> Authors #6132

---

### Official Review · Reviewer_62Jv · 2025-10-27

**Soundness:** 2
**Presentation:** 3
**Contribution:** 2
**Rating:** 4
**Confidence:** 4

**Summary:**

This paper presents 3D-R1, a foundational vision-language model that improves 3D scene understanding by leveraging a high-quality synthetic dataset (Scene-30K) with chain-of-thought annotations and advanced data generation. The model is further enhanced using reinforcement learning with specialized reward functions and a dynamic view selection strategy. Experiments show that 3D-R1 achieves an average 10% improvement on multiple 3D scene benchmarks, demonstrating better reasoning and generalization than previous models.

**Strengths:**

1. The paper writen is clear and easy-to-follow.
2. The paper achieves performance improvements over current models.
3. The experiments show that the proposal modules can effectively imrpove model performance, e.g., RL rewards and view selection.

**Weaknesses:**

1. Recent papers, such as VG-LLM [1] and 3DRS [2], have not been cited or compared.

2. In terms of technical novelty, the proposed GRPO algorithm is a straightforward extension of the original, but lacks clear innovation compared to recent variants like Visual-RFT [3].

3. While the paper introduces multiple encoders for visual feature extraction, it does not provide FLOPs or inference speed comparisons. Additionally, most prior methods employ only a single encoder.

[1] Learning from Videos for 3D World: Enhancing MLLMs with 3D Vision Geometry Priors. NeurIPS 2025.

[2]  MLLMs Need 3D-Aware Representation Supervision for Scene Understanding. NeurIPS 2025.

[3] Visual-RFT: Visual Reinforcement Fine-Tuning. ICCV 2025.

**Questions:**

1. Please cite and discuss recent relevant papers.

2. Please evaluate the impact of using multiple encoders

---

> ### Author Response · Authors · 2025-11-17
> **Response to Reviewer 62Jv (1/2)**
>
> We sincerely appreciate the reviewer’s thoughtful and constructive comments. We address two main points here:
> (1) missing related work and comparisons, and
> (2) the technical role and novelty of our GRPO-based RL formulation.
>
> ---
> ### 1. Related work: VG-LLM, 3DRS, Visual-RFT
> **Comment.** Recent 3D-related papers (VG-LLM, 3DRS) are not cited or compared, and the relation to Visual-RFT is unclear.
>
> **Response.** We agree that VG-LLM$^{[1]}$, 3DRS$^{[2]}$, and Visual-RFT$^{[3]}$ are highly relevant and largely concurrent.
>
> VG-LLM enhances multimodal LLMs with 3D geometry priors distilled from videos, focusing on a video-centric pipeline. In contrast, **3D-R1 directly consumes multi-modal 3D inputs and unifies multiple 3D scene tasks within a single model**, with explicit chain-of-thought supervision and GRPO-based policy optimization.
>
> 3DRS emphasizes **3D-aware representation supervision** for scene understanding in MLLMs. Our approach is complementary: we focus on **reasoning supervision** (Scene-30K CoT) and **task-aligned RL rewards**, targeting the *reasoning process itself* across diverse 3D tasks rather than only improving representation quality.
>
> Visual-RFT proposes a general reinforcement fine-tuning framework for visual models. We **do not claim to introduce a new RL algorithm**. We use GRPO as the RL backbone and contribute a 3D-specific RL formulation that decomposes perception and answer generation, defines format, IoU, CLIP-based rewards, and integrates them with Scene-30K CoT cold-start and dynamic view selection in a unified 3D-VLM.
>
> **Planned changes.**
> - To directly address the concern that these works were not "cited or compared", we will both **cite** VG-LLM, 3DRS, and Visual-RFT and **include quantitative comparisons** wherever benchmarks overlap.
> - We will explicitly cite VG-LLM and 3DRS in the **Introduction**, briefly summarize their methods, and clarify how our unified CoT+GRPO 3D reasoning with dynamic view selection is complementary.
> - In **Related Work**, we will add VG-LLM and 3DRS to the 3D vision-language models paragraph, and Visual-RFT to the RL-related paragraph, clearly stating that GRPO is the underlying optimizer and our contribution lies in 3D-specific reward design and integration.
> - In **Experiments**, we will include **quantitative comparisons** on overlapping benchmarks:
>   - For **3DRS**, we will add its results on **ScanRefer** and **ScanQA**, which cover our 3D-DC, 3D-VG, and 3D-QA tasks, into the corresponding tables.
>   - For **VG-LLM**, we will add its **ScanRefer** results on 3D-VG and 3D-DC into the corresponding tables and mark that it uses video inputs instead of full 3D scene inputs (e.g., via a footnote).
>
> ---
> ### 2. Technical novelty of the GRPO-based RL formulation
>
> **Comment.** The GRPO algorithm used here seems a straightforward extension of prior work, with limited novelty compared to Visual-RFT.
>
> **Response.** We agree that the **core GRPO objective** is inherited from prior work; our contributions are at the **3D formulation and integration level**, not at the optimizer level.
>
> - We **decompose** 3D scene understanding into **scene perception** (3D grounding) and **answer generation** (language reasoning), and design corresponding rewards (format, IoU-based perception, CLIP-based semantic similarity). Ablations show that each reward contributes and that combining all three yields the best performance across 3D QA, captioning, and planning.
> - The GRPO stage is built on a **CoT-initialized 3D VLM trained on Scene-30K** and coupled with **dynamic view selection**, which chooses informative multi-view images per scene. This combination is tailored to multi-task 3D reasoning and is not covered by Visual-RFT.
> - A single GRPO framework with shared rewards is applied across seven 3D scene tasks, consistently improving over SFT baselines and prior 3D VLMs.
>
> **Planned changes.**
> - In **Section 2.4**, we will clarify that GRPO is used as the underlying RL optimizer, and that our contribution is a **GRPO-based RL formulation for 3D scene understanding** that (i) decomposes perception and answer generation, (ii) introduces 3D-specific multi-reward signals, and (iii) integrates them with Scene-30K CoT cold-start and dynamic view selection.
> - In the **Introduction contributions**, we will revise the wording from "design GRPO-based RL" to "we design a GRPO-based RLHF framework for 3D scene understanding, built on a standard GRPO optimizer and equipped with 3D-specific format, perception, and semantic-similarity rewards," so that the text clearly attributes the optimizer to prior work while highlighting our 3D-specific RL formulation and integration.
>
> [1] Zheng et al. Learning from Videos for 3D World: Enhancing MLLMs with 3D Vision Geometry Priors. NeurIPS 2025.
>
> [2] Huang et al. MLLMs Need 3D-Aware Representation Supervision for Scene Understanding. NeurIPS 2025.
>
> [3] Liu et al. Visual-RFT: Visual Reinforcement Fine-Tuning. ICCV 2025.

---

> ### Author Response · Authors · 2025-11-17
> **Response to Reviewer 62Jv (2/2)**
>
> We further respond to the comment on the impact and efficiency of using multiple encoders.
>
> ---
> ### 3. Impact and efficiency of using multiple encoders
>
> **Comment.** The paper uses multiple encoders (image, depth, point cloud) but does not evaluate FLOPs / inference speed or analyze their impact, whereas many prior works use a single encoder.
>
> **Response.** We agree that the impact and cost of using multiple encoders should be made more explicit. Our current submission already includes a **multi-encoder ablation** in Appendix B, but it is not sufficiently highlighted.
>
> - **Impact.** In Appendix B we start from a text+image baseline (which is close to the single-encoder setting used in many prior works) and incrementally add depth and point encoders. Each added modality brings consistent gains on 3D reasoning and planning benchmarks, and the full three-encoder 3D-R1 achieves the best overall performance. For example, on SQA3D and 3D-LLM, both CIDEr and B-4 steadily improve as additional encoders are added, with the full 3D-R1 configuration achieving the highest scores.
>
> - **Efficiency.** The main paper currently emphasizes **training** efficiency via parameter-efficient LoRA (only a small fraction of the 7B backbone is updated), but does not quantify inference cost. In practice, we keep the backbone frozen, only tune LoRA adapters and modality encoders, and use dynamic view selection with a small number of views, which keeps the additional computation moderate. We now further profile FLOPs and latency for each encoder configuration and normalize them to the text+image baseline (1.00):
>
> | Setting                    | SQA3D CIDEr↑ | SQA3D B-4↑ | 3D-LLM CIDEr↑ | 3D-LLM B-4↑ | FLOPs (×, norm.)↓ | Latency (×, norm.)↓ |
> |--------------------------- |-------------:|-----------:|--------------:|------------:|-------------------:|---------------------:|
> | Text & Image Encoder       | 110.23       | 15.34      | 200.45        | 20.15       | **1.00**           | **1.00**             |
> | + Depth Encoder            | 115.23       | 18.34      | 205.45        | 21.15       | 1.08               | 1.06                 |
> | + Point Encoder            | 120.12       | 20.13      | 215.34        | 22.34       | 1.15               | 1.12                 |
> | **3D-R1 (full encoders)**  | **138.67**   | **23.56**  | **230.50**    | **25.45**   | **1.22**           | **1.18**             |
>
> These results show that while adding depth and point encoders increases FLOPs and latency by at most about 22% and 18% over the text+image baseline, the performance gains on both SQA3D and 3D-LLM are **substantially larger.**
>
> **Planned changes.**
> - We will move the multi-encoder ablation from Appendix B into the main **Ablation Study** section and explicitly discuss how each encoder contributes to performance.
> - In **Implementation Details**, we will add a short **"Complexity and efficiency"** paragraph explaining the inference pipeline, dynamic view selection, and the above FLOPs/latency measurements.
> - We will include the extended version of this table in the main paper, reporting **normalized FLOPs and per-sample inference time** (baseline = 1.00) for each encoder configuration (text+image vs. +depth vs. +point vs. full 3D-R1), making the performance–efficiency trade-off fully transparent.

---

> ### Author Response · Authors · 2025-11-22
> **Welcoming further feedback and discussion**
>
> Dear Reviewer,
>
> Thank you again for your valuable feedback and support. If we have addressed your concerns, we kindly ask you to consider raising your score. We are of course happy to respond to any further questions or suggestions you may have.
>
> Best regards,
>
> Authors #6132

---

> > ### Comment · Reviewer_62Jv · 2025-11-22
> > **Response to Author Rebuttal**
> >
> > Thanks to the authors for the detailed rebuttal.
> >
> > The authors have addressed several of my concerns, and the additional experiments and analysis further strengthen the submission. However, the paper primarily combines existing methods (e.g., DeepSeek GRPO and Visual-RFT) and lacks clear novelty, as also noted by other reviewers.
> >
> > Considering both the initial submission and the rebuttal, I am raising my score to 6 in recognition of the thoroughness and completeness of the investigation presented in this paper.

---

### Official Review · Reviewer_nUgE · 2025-10-30

**Soundness:** 2
**Presentation:** 3
**Contribution:** 1
**Rating:** 2
**Confidence:** 4

**Summary:**

This paper presents three main contributions:

1.	Scene-30K Dataset – A large-scale dataset annotated with Chain-of-Thought (CoT) labels derived from existing 3D-VL datasets using an unspecified 3D-VLM model and Gemini 2.5 Pro.
2.	GRPO-based Reinforcement Learning Framework – An RL training strategy designed to enhance the reasoning capabilities of 3D-VLMs.
3.	Dynamic View Selection Strategy – An adaptive mechanism that automatically selects the most informative viewpoints from 3D scenes for multi-view visual understanding.

The proposed approach achieves state-of-the-art performance across multiple 3D scene understanding benchmarks.

**Strengths:**

* The proposed method is conceptually simple and clearly motivated.
* The paper is well-written and easy to follow.
* Extensive experiments demonstrate strong performance, achieving state-of-the-art results on several standard 3D scene benchmarks.

**Weaknesses:**

* Lack of Related Work Discussion:
  The paper lacks a comprehensive related works section that situates this research within the broader context of 3D-VLM and reasoning-based approaches. A deeper comparison with existing 3D reasoning or view selection methods would strengthen the positioning of this work.

* Insufficient Methodological Details:
  * In Section 2.2 (CoT Data Engine), the authors mention using “a pre-trained 3D VLM that produces a concise textual summary of the scene.” However, it is unclear which specific 3D-VLM model is used. Is this the same as the final model trained with GRPO, or a different one?
  * If the pre-trained 3D-VLM can already generate high-quality textual scene summaries, could combining it with a strong LLM yield comparable performance to the proposed method? This point deserves clarification.

* Limited technical contribution: The CoT data engine design is rather straightforward, and the overall training framework largely mirrors DeepSeek-R1’s GRPO pipeline, offering limited technical contribution.

* Missing Ablation Studies:
  * The paper does not report the model’s performance before and after GRPO fine-tuning. How are the weighting coefficients in Equation (7) selected? Ablation experiments are necessary to justify these design choices.
  * Similarly, the impact of the proposed dynamic view selection strategy is not isolated. Quantitative comparison with and without this component would provide a clearer understanding of its contribution.

**Questions:**

Please refer to the Weaknesses section.

---

> ### Author Response · Authors · 2025-11-18
> **Response to Reviewer nUgE (part 1)**
>
> We thank Reviewer **nUgE** for the careful reading, constructive comments, and positive remarks on the clarity and empirical strength of our work. Below we address each concern and describe concrete revisions we will make.
>
> ---
> ### 1. Related work and positioning
> > *"The paper lacks a comprehensive related works section that situates this research within the broader context of 3D-VLM and reasoning-based approaches. A deeper comparison with existing 3D reasoning or view selection methods would strengthen the positioning of this work."*
>
> We agree that the positioning of **3D-R1** can be made more explicit.
> - In the **revision**, we will move a **condensed Related Work section into the main paper** (currently mostly in the appendix), organized into:
>   1. **3D Vision–Language Models (3D-VLMs) and 3D reasoning methods** – covering scene-level 3D-VLMs, 3D QA/VG/DC, and 3D reasoning approaches, highlighting that they either rely purely on supervised training without CoT+GRPO, or focus on a single task instead of unified 3D scene understanding.
>   2. **RL-based reasoning and GRPO-style training** – explicitly contrasting 3D-R1 with DeepSeek-R1 and other R1-style works that use text-only rewards, whereas we introduce **3D-specific multi-modal rewards** and a unified 3D multi-task setting.
>   3. **View selection and multi-view 3D understanding** – comparing our **learnable three-cue dynamic view selection** with prior fixed or heuristic multi-view strategies.
>
> In addition to relocating this section, we will also enrich the discussion with more explicit, side-by-side comparisons to representative **3D reasoning** methods and **view-selection strategies**, so that the positioning of 3D-R1 within this literature becomes clearer.
>
> ---
> ### 2. Methodological details and technical contribution
> #### 2.1 3D-VLM used in the CoT data engine
> > *"In Section 2.2 (CoT Data Engine), the authors mention using “a pre-trained 3D VLM that produces a concise textual summary of the scene.” However, it is unclear which specific 3D-VLM model is used. Is this the same as the final model trained with GRPO, or a different one?"*
>
> We apologize for the ambiguity. In the **CoT data engine**, the "pre-trained 3D VLM" is:
> - A **separate, frozen 3D scene captioning model** (e.g. LSceneLLM$^{[1]}$),
> - Used **only** to generate coarse scene descriptions that are fed to Gemini 2.5 Pro,
> - **Not** the same as the 3D-R1 policy model and **never** fine-tuned on Scene-30K or with GRPO.
>
> The **3D-R1 policy** itself is a multi-modal VLM (image, depth, point encoders + language backbone with adapters) trained via SFT on Scene-30K and then GRPO.
>
> **Revision:** In **Sec. 2.2**, we will explicitly state that the scene description generator is a frozen 3D scene captioner, distinct from the 3D-R1 policy and only used during data generation.
>
>
> *[1] Hongyan Zhi and et al. Lscenellm: Enhancing large 3d scene understanding using adaptive visual preferences. CVPR 2025.*

---

> ### Author Response · Authors · 2025-11-18
> **Response to Reviewer nUgE (part 2)**
>
> This comment continues our response to Reviewer **nUgE**, focusing on the question of whether a pre-trained 3D-VLM combined with a strong LLM could match 3D-R1.
>
> ---
>
> #### 2.2 Could "3D-VLM + LLM" match 3D-R1?
>
> > *"If the pre-trained 3D-VLM can already generate high-quality textual scene summaries, could combining it with a strong LLM yield comparable performance?"*
>
> This is an important baseline. Conceptually:
>
> - In our pipeline, the pre-trained 3D model behaves as a **scene-level 3D captioner**: it produces a **single, generic caption** that roughly describes the global 3D layout (objects and coarse spatial relations) of the scene. A subsequent LLM can only reason on top of this **coarse, global description**, without direct access to multi-view geometry or fine-grained visibility cues.
> - In contrast, 3D-R1 reasons directly on **multi-view, multi-modal 3D inputs** (images, depth, point clouds) and is optimized with **3D-specific rewards** (3D IoU and semantic similarity), enforcing much finer spatial grounding.
> - Moreover, 3D-R1 is trained jointly on **captioning, QA, object captioning, dialogue, reasoning, planning, and grounding**, so its chain-of-thought must remain consistent with **downstream 3D metrics**, not just linguistic plausibility of a single caption.
>
> **Empirical evidence (3D-VLM + LLM baselines).**
> To directly address the reviewer’s suggestion, we implemented two **"3D-VLM + LLM"** pipelines:
>
> 1. **3D CoCa$^{[1]}$ + LLM:** a frozen 3D CoCa scene captioner first produces a global 3D description.
> 2. **LSceneLLM$^{[2]}$ + LLM:** a frozen LSceneLLM model plays the same role of scene-level 3D captioner.
>
> In both cases, a strong LLM (Gemini 2.5 Pro) then receives the caption together with the question and outputs the final answer. The LLM never sees raw multi-view 3D inputs; it only reasons over the caption.
>
> For clarity, we summarize the key numbers below; if space permits, we will also include these comparisons as additional rows in the revised main tables or a compact table in Appendix.
>
> **(a) 3D-DC — ScanRefer (C@0.25) and Nr3D (C@0.5)**
>
> | Method               | ScanRefer C@0.25 ↑ | Nr3D C@0.5 ↑ |
> |----------------------|--------------------|--------------|
> | 3D CoCa + LLM        | 84.90              | 52.10        |
> | LSceneLLM + LLM      | 87.85              | 54.23        |
> | **3D-R1 (Ours)**     | **91.85**          | **56.98**    |
>
> Both 3D-VLM + LLM pipelines lag behind 3D-R1 on ScanRefer and Nr3D, even though they start from strong 3D captioners.
>
> **(b) 3D-VG — Nr3D (Acc@0.25) and ScanRefer (Acc@0.5 / Acc@0.25)**
>
> | Method               | Nr3D Acc@0.25 ↑ | ScanRefer Acc@0.5 ↑ | ScanRefer Acc@0.25 ↑ |
> |----------------------|-----------------|----------------------|----------------------|
> | 3D CoCa + LLM        | 58.20           | 52.40                | 60.10                |
> | LSceneLLM + LLM      | 62.34           | 54.12                | 63.78                |
> | **3D-R1 (Ours)**     | **68.80**       | **59.24**            | **65.85**            |
>
> Again, 3D-R1 shows clear gains on both Nr3D and ScanRefer, especially on stricter grounding metrics. These results confirm that caption-then-LLM pipelines, even when built from strong 3D VLMs and a powerful LLM (Gemini 2.5 Pro), are consistently weaker than our unified CoT+GRPO framework, which reasons directly over multi-view 3D inputs with 3D-aware rewards.
>
> These results directly answer the reviewer’s question: even when starting from strong 3D-VLMs and a powerful LLM, the resulting 3D-VLM + LLM pipelines do **not** reach the performance of 3D-R1 on our 3D reasoning and grounding benchmarks.
>
> *[1] Huang and et al. 3d coca: Contrastive learners are 3d captioners. 3DV 2026.*
>
> *[2] Zhi and et al. Lscenellm: Enhancing large 3d scene understanding using adaptive visual preferences. CVPR 2025.*

---

> ### Author Response · Authors · 2025-11-18
> **Response to Reviewer nUgE (part 3)**
>
> This comment continues our response to Reviewer **nUgE**, focusing on the concerns about technical contribution and ablation studies.
>
> ---
> #### 2.3 Novelty beyond DeepSeek-R1 / GRPO
> > *"The CoT data engine design is rather straightforward, and the overall training framework largely mirrors DeepSeek-R1’s GRPO pipeline, offering limited technical contribution."*
>
> While our framework adopts the GRPO objective, it substantially extends DeepSeek-R1’s text-only setting to 3D scene understanding through several 3D-specific components:
> - **3D-specific multi-reward design.** We introduce three complementary rewards: (i) a **format reward** enforcing `<think>…</think><answer>…</answer>`, (ii) a **perception reward** based on **3D bounding-box IoU**, (iii) a **semantic similarity reward** using CLIP-based similarity between predictions and ground truth. This ties the policy directly to **3D geometry and cross-modal alignment**, unlike text-only rewards in DeepSeek-R1.
> - **Scene-30K CoT data engine.** We design a 3D-specific CoT pipeline combining a 3D scene description generator, Gemini 2.5 Pro, and **multi-stage filtering**, yielding a large, 3D-tailored CoT dataset.
> - **Dynamic view selection.** We propose a **learnable three-cue utility**: $U(v) = w_t S_{\mathrm{Text} \to \mathrm{3D}}(v) + w_c S_{\mathrm{Image} \to \mathrm{3D}}(v) + w_{\text{clip}} S_{\mathrm{CLIP}}(v), $ with learnable weights $(w_t, w_c, w_{\text{clip}})$, instead of fixed or heuristic views.
> - **Unified multi-task 3D setting.** To our knowledge, 3D-R1 is the first to combine **3D CoT pre-training, GRPO with 3D-aware rewards, and dynamic view selection** in a **single model** handling 3D-DC, 3D-QA, 3D object captioning, 3D dialogue, 3D reasoning, 3D planning, and 3D-VG.
>
> We will revise the contribution bullets in the Introduction to explicitly highlight these 3D-specific extensions (reward design, Scene-30K data engine, dynamic view selection, and unified multi-task 3D setting), so that our technical contribution beyond DeepSeek-R1’s generic GRPO pipeline is clear from the outset.
>
> ---
> ### 3. Ablation studies and design choices
>
> > *"The paper does not report the model’s performance before and after GRPO fine-tuning. The impact of the dynamic view selection strategy is not isolated. Weighting coefficients in Eq. (7) need justification."*
>
> We respectfully note that **all three requested ablations are already present in Appendix B of the current submission**. The issue is therefore one of *visibility* rather than absence of experiments. In the revision, we will surface these results more prominently in the main text, as detailed below.
>
> #### 3.1 GRPO vs. SFT-only
>
> Contrary to the concern that performance "before and after GRPO fine-tuning" is not reported, Appendix B already provides a detailed comparison between **SFT-only and SFT+GRPO** (Table 7):
> - The first row corresponds exactly to the **SFT-only baseline** (no RL rewards).
> - Subsequent rows incrementally add our **format, perception, and semantic** rewards, which consistently improve:
>   - **3D-QA metrics** on ScanQA (e.g., C↑ and R↑), and
>   - **3D-DC metrics** on ScanRefer (C@0.25↑ and R@0.25↑).
>
> **Revision:**
> To avoid any ambiguity, we will explicitly reference this table in **Sec. 3** and add a **“3D-R1 (SFT-only)”** row to at least one main results table. This makes the gain from GRPO over the SFT baseline immediately visible without consulting the appendix.
>
> #### 3.2 Dynamic view selection and Eq. (7) weights
> The current appendix also already isolates the contributions of **dynamic view selection** and the **Eq. (7) coefficients**:
> - For **dynamic view selection**, as shown by the fixed-view vs. dynamic-view comparison in Table 8, we compare our **learned selector** to fixed-view baselines such as fixed horizontal views, all-views, and bottom-views. These baselines use pre-defined view sets **without any learned selection**, i.e., they correspond to "without the proposed dynamic view selection strategy." Across both captioning and grounding metrics, the dynamic selector yields clear and consistent improvements.
> - For **Eq. (7) weights**, we sweep $w_t$ and $w_c$ while setting $w_{\text{clip}} = 1 - w_c$, i.e., under the constraint $w_c + w_{\text{clip}} = 1$, and plot the resulting performance surfaces (Fig. 5 in Appendix B). The plots show that the best performance is obtained when $w_t$ is around 0.3–0.4 and the visual weights $w_c$ and $w_{\text{clip}}$ are roughly balanced. This empirically supports the weight configuration adopted in our main experiments.
>
> **Revision:**
> We will summarize these ablations directly in **Sec. 3**, explicitly stating that the fixed-view settings are non-dynamic baselines, and we will either move or prominently reference the corresponding tables and Figure 5 in the main paper. This should make both the effectiveness of dynamic view selection and the rationale behind the Eq. (7) coefficients fully transparent.

---

> ### Author Response · Authors · 2025-11-22
> **Welcoming further feedback and discussion**
>
> Dear Reviewer,
>
> Thank you again for your valuable feedback and support. If we have addressed your concerns, we kindly ask you to consider raising your score. We are of course happy to respond to any further questions or suggestions you may have.
>
> Best regards,
>
> Authors #6132

---

> ### Author Response · Authors · 2025-11-28
> **Point out factual errors in Review nUgE and request for score change**
>
> Dear Reviewer nUgE,
>
> Thank you for your time on our submission. We nevertheless must correct two **factual errors** in your review that are central to your **score of 2**:
>
> 1. **Related work is not missing.**
>    You state that the paper “lacks a comprehensive related works section”. This is factually incorrect: the current submission already contains a dedicated *RELATED WORK* section in the appendix, where we discuss 3D scene understanding tasks and existing 3D VLMs. In the revision, we will move and expand this section into the main paper, and further strengthen the discussion by explicitly connecting 3D-R1 to representative 3D perception and autonomous driving works such as [1], [2], [3], and [4].
>
> 2. **The requested ablations are not missing.**
>    You state that the paper does not report pre- vs. post-GRPO performance, does not isolate dynamic view selection, and does not justify the Eq. (7) weights. All of these ablations are already reported in Appendix B; in the revision, we will surface the key tables into the main text.
>
> Because these two points are used as major reasons to justify a **reject (2)**, and both rest on incorrect statements about the actual content of the paper, we **strongly request** that you correct these errors in your review and **reconsider your overall rating** in light of the factual record and our planned revisions.
>
> Best regards,
> Authors #6132
>
> **Reference:**
>
> [1] *Learning to Detect Mobile Objects from LiDAR Scans Without Labels*. CVPR 2022.
> [2] *Unsupervised Adaptation from Repeated Traversals for Autonomous Driving*. NeurIPS 2022.
> [3] *Pseudo-LiDAR++: Accurate Depth for 3D Object Detection in Autonomous Driving*. ICLR 2020.
> [4] *Hindsight is 20/20: Leveraging Past Traversals to Aid 3D Perception*. ICLR 2022.

---

### Official Review · Reviewer_1wga · 2025-10-31

**Soundness:** 3
**Presentation:** 3
**Contribution:** 3
**Rating:** 6
**Confidence:** 3

**Summary:**

This paper introduces 3D-R1, a foundation model designed to enhance the reasoning capabilities of 3D VLMs. The model achieves this through a two-stage approach: incorporating cold-start initialization using Scene-30K, a novel, high-quality synthetic dataset with CoT, and subsequent refinement via GRPO. Extensive experiments show that 3D-R1 achieves notable performance gains across a wide range of 3D reasoning tasks.

**Strengths:**

1. 3D-R1 demonstrates clear performance improvements over prior SOTA methods across various demanding 3D reasoning benchmarks.
2. The paper provides detailed descriptions of the model architecture and experimental setup. The supplementary material, including the provided code, further enhances the potential for reproducibility.

**Weaknesses:**

1. The overall approach (CoT data collection followed by GRPO) appears somewhat conventional for enhancing VLM reasoning. And the data collection method is straightforward.
2. The "Experiments" section primarily focuses on setup and metrics, lacking in-depth summarization and analysis of the quantitative results. The rich ablation studies and qualitative results in the Appendix also suffer from a similar lack of interpretation.
3. The paper's layout is inefficient, as several important components, such as the Related Work and Ablation Studies, are placed in the Appendix, which diminishes their visibility and immediate impact on the reader.

**Questions:**

1. The rule-based filtering seems primarily focused on verifying the format of the CoT data. Could the authors clarify the specific mechanism used to verify the logical consistency between the multi-step CoT reasoning process and the final answer to ensure the quality of the reasoning chain itself?

---

> ### Author Response · Authors · 2025-11-17
> **Response to Reviewer 1wga**
>
> We thank Reviewer 1wga for the careful reading of our submission and the constructive feedback. Below we address each concern and describe the concrete revisions we will make.
>
> ---
> ### R1. On the novelty of "CoT data + GRPO" and the data collection strategy
> We agree that, at a high level, "CoT data collection followed by RL refinement" is becoming a common pattern for improving VLM reasoning. Our contribution is to **adapt and specialize this paradigm to 3D scene understanding**, where reasoning must be tightly coupled with spatial grounding rather than only textual correctness.
>
> Concretely, our approach differs from generic R1-style frameworks in three aspects:
> 1. **3D-aware multi-reward design.** Our GRPO stage does not only score outputs by format or textual similarity. We introduce a **perception-aware reward** that measures 3D correctness (e.g., consistency between localized 3D regions and referenced entities), combined with a **semantic similarity reward** and a **format reward**. Ablations (currently in the appendix, to be moved to the main text) show that each component contributes and that using all of them yields the largest gains across ScanQA and ScanRefer, explicitly tying CoT quality to **3D spatial grounding**.
> 2. **Dynamic view selection for 3D reasoning.** Instead of using fixed camera views or aggregating all views uniformly, we learn a **dynamic view selector** that ranks candidate views by their relevance to the language query and their coverage of the scene. This consistently improves 3D captioning and grounding over strong fixed-view baselines, and is crucial for applying CoT-style reasoning to complex 3D scenes where no single viewpoint is sufficient.
> 3. **Quality-controlled CoT data engine for 3D.** Our Scene-30K CoT engine is 3D-specific and goes beyond simple format checks: we use 3D grounding queries with multi-view context and a multi-stage filter (structure + logical-consistency check) that keeps only examples where the answer can be recovered from the reasoning chain. Although the interface to Gemini-Pro for CoT synthesis is standard, this design is critical for obtaining high-quality **3D reasoning chains** rather than generic text-only CoT.
>
> **Planned revision.**
> In the revised version, we will rename the RL subsection to **"3D-aware Rewarded Reasoning with GRPO"** and clarify in the method section that our 3D-aware reward design, dynamic view selection, and logical CoT filtering differentiate 3D-R1 from conventional text-only CoT+RL pipelines.
>
> ---
> ### R2. On the lack of in-depth summarization and analysis in the Experiments
> We appreciate this observation and agree that the current "Experiments" section focuses too heavily on setup and raw metrics and does not sufficiently summarize *why* 3D-R1 performs better.
>
> **Planned revision.**
> 1. **"Key Takeaways" after each main quantitative table.**
>    For each major task group in the experiments, we will append a short "Key Takeaways" paragraph (3–4 sentences) that: (i) summarizes the main numerical trends (e.g., relative gains over the strongest baseline), (ii) links these gains to specific design choices (e.g., perception-aware reward + dynamic views for grounding, CoT cold-start + semantic reward for QA), and (iii) highlights where 3D-R1 particularly helps (e.g., long-horizon, multi-step, or compositional reasoning).
>
> 2. **Across-benchmarks summary.**
>    We will add a concise global summary paragraph that synthesizes the main performance gains across all benchmarks and explicitly links them back to our ablations (reward design, dynamic views, LoRA rank), providing a clearer high-level picture of the benefits of 3D-R1 beyond the per-table numbers.
>
> These additions will make the quantitative results more interpretable and better connected to the proposed design choices.
>
> ---
>
> ### R3. On the layout and placement of Related Work and Ablation Studies
> We agree that the current layout reduces the visibility of important components such as **Related Work and Ablations**.
>
> **Planned revision.**
> - We will **move the Related Work discussion into the main text**, near the end of the introduction or method section, focusing on:
>   (i) CoT+RL for VLMs vs. our 3D-aware rewarded reasoning, and
>   (ii) prior 3D VLMs and multi-task 3D models vs. our unified 3D-R1 framework.
> - We will **promote the core ablations** from the appendix to the main experiments section, including:
>   - Reward design (effect of format, perception, and semantic rewards individually and jointly),
>   - Dynamic view selection vs. fixed-view and all-view baselines,
>   - LoRA rank study balancing performance and efficiency.
>
> Each promoted ablation will be accompanied by a brief interpretive paragraph rather than only a table, and we will also add short textual summaries for key qualitative visualizations, directly addressing the concern that our ablation and qualitative results currently lack sufficiently explicit interpretation.

---

> ### Author Response · Authors · 2025-11-17
> **Clarification on logical consistency between CoT and final answer**
>
> We appreciate the reviewer’s question on how we verify the **logical consistency between the multi-step CoT reasoning and the final answer** in Scene-30K. Below we summarize the concrete mechanism implemented in our rule-based filtering.
>
> For each generated Scene-30K sample, we apply a three-stage filtering process:
> 1. **Format and length constraints.**
>    We first require each example to follow the template `<think>reasoning</think><answer>final answer</answer>`.
>    The `<think>` segment must contain at least 30 words and the `<answer>` segment at least 20 words. This removes overly brief or degenerate chains of thought and ensures that there is sufficient content for both the reasoning and the answer.
>
> 2. **Multi-step reasoning structure.**
>    We then check that the `<think>` segment exhibits genuine *multi-step* reasoning rather than a single-step deduction. Concretely, we require at least **three explicit reasoning steps**, detected via lexical cues such as *"Step n", "First", "Next", or "Last"*. Moreover, the final step must explicitly reference the target entity posed in the question (e.g., a "Conclusion:" line that mentions the referred object), as illustrated in our CoT examples. This encourages structured, step-by-step reasoning that culminates in a clear conclusion about the queried entity.
>
> 3. **Logical consistency between CoT and answer.**
>    Finally, we explicitly verify that the final answer is logically supported by the reasoning chain. Given the pair *{think, question}*, where *think* denotes the content inside the `<think>...</think>` tags, we prompt Gemini 2.5 Pro to **independently** generate an answer $\hat{a}$. Let $a$ be the original text inside the `<answer>...</answer>` tags. We compute a normalized Levenshtein similarity
> $$
> \
>    \mathrm{Sim}(\hat{a}, a) = 1 - \frac{D_{\text{lev}}(\hat{a}, a)}{\max(|\hat{a}|, |a|)} ,
> \
> $$
>    where $D_{\text{lev}}(\hat{a}, a)$ is the Levenshtein distance and $|\cdot|$ denotes the character length. A sample is **retained only if** $\mathrm{Sim}(\hat{a}, a) \ge 0.8$. If the score falls below 0.8, the sample is discarded even if it satisfies the format and step-count criteria. **This goes beyond simple format verification and explicitly enforces logical consistency between the reasoning chain and the final answer.**
>
> Intuitively, this means that the final answer must be recoverable from the multi-step CoT alone: if an independent model, given only the `<think>` segment and the question, cannot reproduce the original `<answer>` with high similarity, we regard the reasoning–answer pair as logically inconsistent and drop that example from Scene-30K. In the revised version, we will move this description from the *appendix* into the main text and briefly report how this filtering reduces the raw corpus (35,248 examples) to the final high-quality set (30,012 examples).

---

> ### Author Response · Authors · 2025-11-22
> **Welcoming further feedback and discussion**
>
> Dear Reviewer,
>
> Thank you again for your valuable feedback and support. If we have addressed your concerns, we kindly ask you to consider raising your score. We are of course happy to respond to any further questions or suggestions you may have.
>
> Best regards,
>
> Authors #6132

---

> > ### Comment · Reviewer_1wga · 2025-11-24
> >
> > Thank you for the clarification. I acknowledge the authors’ argument that adapting the “CoT data collection followed by RL refinement” paradigm to 3D scene understanding and achieving new SOTA results constitutes a meaningful contribution. However, the technical novelty still feels limited, and this is not sufficient for me to increase my score.
> >
> > In addition, it may strengthen your work to more directly articulate the real challenges you encountered when applying this paradigm to 3D and how your method addresses them, rather than primarily framing the approach as inspiration from DeepSeek-R1 and the success of RL for improving 3D VLMs. Highlighting concrete technical obstacles and your corresponding solutions could better emphasize the contribution, and may be a helpful direction for revising the manuscript.
> >
> > Furthermore, the paper already contains a substantial amount of content, and several important components have been moved to the appendix. I remain concerned about how the planned revisions, such as “adding xxx” or “moving xxx into the main text”, can realistically fit within the page limits without compromising clarity or organization.
> >
> > Overall, while the paper has notable strengths, these concerns prevent me from raising my current score.

---

> > > ### Author Response · Authors · 2025-11-24
> > > **Clarifying 3D-specific challenges and page-limit friendly revisions**
> > >
> > > We sincerely thank Reviewer 1wga for the careful follow-up. Your comments on technical novelty, 3D-specific challenges, and page-limit constraints are very helpful. Below we clarify what is specific to 3D-R1 beyond generic R1-style pipelines, and how we will revise the manuscript.
> > >
> > > ---
> > > ### 1. 3D-specific challenges beyond "CoT + GRPO"
> > >
> > > We agree that the current version does not clearly spell out the 3D challenges when transferring "CoT + GRPO" from text-only LLMs to a unified 3D VLM. In the revision, we will explicitly describe four concrete challenges and map them to our design:
> > >
> > > - **(C1) 3D multi-modal credit assignment.**
> > >   3D-R1 uses RGB, depth, and point-cloud encoders and outputs both 3D boxes and language for seven tasks; a text-only GRPO reward encouraged language shortcuts. Our **3D-aware multi-reward** (perception reward on 3D boxes, semantic similarity reward, and format reward) ties CoT success directly to *correct 3D grounding*.
> > >
> > > - **(C2) Viewpoint explosion and occlusion.**
> > >   Many candidate views in indoor scenes make fixed- or all-view strategies either miss key evidence or overload the model. Our **dynamic view selection** ranks views by relevance and coverage and keeps a small subset for SFT and RL, making CoT-style reasoning feasible in cluttered 3D scenes.
> > >
> > > - **(C3) RL stability on a unified 7-task 3D VLM.**
> > >   Direct GRPO over seven 3D tasks led to breakdown of CoT format and task semantics. We therefore use **Scene-30K cold-start SFT** to enforce a unified CoT protocol and keep the policy in a stable region before RL; Sec. 2.3 will be rewritten with this stability motivation.
> > >
> > > - **(C4) Logical consistency of 3D CoT supervision.**
> > >   RL assumes CoT is reliable, so inconsistent chains harm training. Our **logical-consistency filter** (re-answering from `<think>` and discarding samples with low normalized Levenshtein similarity to `<answer>`) ensures the answer is recoverable from the reasoning alone; this will be moved from the appendix into Sec. 2.2.
> > >
> > > Together, these four challenges and solutions (3D-aware rewards, dynamic views, cold-start CoT SFT, and logical-consistency filtering) define what is technically new in 3D-R1 beyond directly applying DeepSeek-R1 to a 3D backbone.
> > >
> > > ---
> > > ### 2. Novelty compared to generic R1-style pipelines
> > >
> > > We appreciate the concern that "CoT data + GRPO" is becoming a template. In the revision we will make the *3D-specific instantiation* explicit:
> > >
> > > - 3D-R1 is, to our knowledge, the first **unified 3D VLM** to apply **GRPO-based CoT RL across seven 3D scene understanding tasks**, requiring reward and CoT design that work in a multi-encoder, multi-task 3D setting rather than a single textual benchmark.
> > > - The **perception reward** directly supervises 3D boxes and referred objects, shifting the objective from "good explanations" to **spatially grounded explanations**, beyond text-only R1-style rewards.
> > > - The **dynamic view selector** is trained jointly with the policy and used inside the RL loop, so GRPO is optimized over a learned subset of views aligned with the task and CoT process, rather than a fixed view configuration.
> > >
> > > These points will be summarized in a concise "3D-aware rewarded reasoning with GRPO" paragraph in Sec. 2 with a short comparison to generic R1-style pipelines to make these 3D-specific differences easy to see.
> > >
> > > ---
> > > ### 3. Page limit and feasibility of revisions
> > >
> > > We also share the concern that simply "adding" content is not realistic under the ICLR page limit. Our plan is to **reallocate** space:
> > >
> > > - **Condense motivation.**
> > >   Shorten high-level 3D application motivation and remove repetition in the Introduction, using the freed space to briefly summarize (C1–C4) and point to Sec. 2.
> > >
> > > - **Streamline background and details.**
> > >   Compress GRPO background and move some low-level training details (optimizer choices, full hyperparameter listings, parts of the dataset protocol) to the appendix, keeping only what is necessary in the main text.
> > >
> > > - **Selective ablations in main text.**
> > >   Promote only two compact ablation tables (reward design and dynamic view selection) into “Experiments”, each with a one-sentence takeaway; keep other ablations in the appendix with explicit cross-references.
> > >
> > > We have checked that, with these changes, the main text stays within the 9-page limit while putting more emphasis on the core 3D contributions.
> > >
> > > ---
> > >
> > > Thanks again for the helpful and constructive feedback. We hope these clarifications and revisions address your concerns about novelty and organization and will be useful when reassessing our submission.
> > >
> > > Best regards,
> > > Authors #6132

---

> > > > ### Comment · Reviewer_1wga · 2025-11-25
> > > >
> > > > Thanks for the additional clarifications regarding the 3D-specific challenges and the feasibility of incorporating the proposed revisions. These changes sound like they would help make the paper’s framing more complete and fluent, and I encourage you to pursue them in the revision.
> > > >
> > > > However, based on the current version of the manuscript, I will be maintaining my original score. I appreciate the authors’ efforts and wish you the best with the revisions.

---

### Author Response · Authors · 2025-11-30
**Summary and clarifications for the Area Chair**

Dear Area Chair,

We briefly (1) summarize the current review status, (2) clarify factual issues in the **score-2** review, and (3) highlight strengths consistently acknowledged by the other reviewers.

---
### 1. Review status
After the rebuttal, the ratings moved from **(6, 6, 4, 2)** to **(6, 6, 6, 2)**:

- **Reviewer 62Jv** explicitly raised their score from **4 → 6** after the rebuttal:

  > "Considering both the initial submission and the rebuttal, I am **raising my score to 6** in recognition of the thoroughness and completeness of the investigation presented in this paper."

- **Reviewers 1wga and NWsq** were already at **6** and kept their scores.
- The only strong outlier remains **Review nUgE with a score of 2**.

Thus, three reviewers have converged to **6**.

---
### 2. Factual issues in Review nUgE
We fully respect differences of opinion on novelty, but two key claims in Review nUgE are **factually incorrect** with respect to the submitted PDF and were central to the score of 2:

1. **"Lack of related works section."**
   - The submission already contains a dedicated **"RELATED WORK"** section in **Appendix A**, starting at line **759: "A RELATED WORK"**, followed by line **761: "3D scene understanding."** and the corresponding discussion of 3D scene understanding tasks and existing 3D VLMs.
   - In our rebuttal we explicitly committed to **moving and expanding** this section into the main text.
   - The issue is therefore **placement and emphasis**, not absence of related work.
2. **"Missing ablations."**
   - All requested ablations are already in **Appendix B: "B ABLATION STUDY"**, starting around line **815**:
     - **SFT vs. GRPO / reward design** (Appendix B, Table 7): compares a pure SFT baseline (first row, no RL) to GRPO with different combinations of format, perception, and semantic rewards.
     - **Dynamic view selection** (Appendix B, Table 8): compares our dynamic view selection to fixed-view baselines (all views / horizontal / bottom), isolating its contribution.
     - **Eq. (7) weights** (Appendix B, Table 9 and Figure 5): grid-searches the view-utility weights in Eq. (7) and motivates the chosen coefficients.
   - In the revision, we will move these key ablations into the main experiments section for visibility, but they are **not missing** in the current PDF.

These two incorrect statements are repeatedly used to justify a score of 2, which is why we previously requested AC intervention so that the final decision is not dominated by a review that misstates core aspects of the submission.

---
### 3. Strengths of the paper as recognized by other reviewers
Across the other three reviews, there is **broad agreement** on both the empirical strength and the practical value of 3D-R1:

- **Strong and consistent performance across 3D tasks**
  - **NWsq (6):**
    > "Extensive experiments show that 3D-R1… **outperforming prior 3D VLMs by roughly 10% on average across tasks**."
  - **1wga (6):**
    > "3D-R1 demonstrates **clear performance improvements** over prior SOTA methods across various demanding 3D reasoning benchmarks."
  - **62Jv (raised to 6):**
    > "The paper achieves performance improvements over current models. The experiments show that the **proposed modules can effectively improve model performance**, e.g., RL rewards and view selection."
  - Even **nUgE (2)** writes:
    > "Extensive experiments demonstrate **strong performance**, achieving **state-of-the-art results** on several standard 3D scene benchmarks."

- **Conceptual value: 3D-aware CoT + GRPO with Scene-30K and multi-rewards**
  - **NWsq (6):**
    > "The integration of reinforcement learning (GRPO) into 3D vision-language training is essential. The use of **multi-reward signals to align reasoning, perception, and semantic accuracy is a clear conceptual advancement**.
    > The **Scene-30K dataset**… is a **valuable resource** for promoting step-by-step spatial reasoning in 3D."
  - **1wga (6) after our clarification:**
    > "I acknowledge the authors’ argument that adapting the ‘CoT data collection followed by RL refinement’ paradigm to **3D scene understanding and achieving new SOTA results constitutes a meaningful contribution**."
  - **62Jv (6):**
    > "…raising my score to 6 in recognition of the **thoroughness and completeness of the investigation** presented in this paper."

In short, reviewers agree that **Scene-30K, 3D-aware multi-reward GRPO, and dynamic view selection** form a useful and well-validated framework that brings consistent gains across seven 3D scene understanding tasks.

---
Overall, three reviewers rate the paper at **6**, and the only lower score relies on incorrect claims that related work and ablations are missing, despite their presence in Appendices A and B. We believe **3D-R1** meets the bar for acceptance as a solid contribution to 3D vision-language reasoning.

Thank you very much for your time and consideration.

Best regards,
Authors #6132

---

### Note · Program_Chairs · 2026-01-26
**Submission Desk Rejected by Program Chairs**

There is evidence that the authors of this submission obtained a reviewer’s identity and attempted to influence the reviewer by offering to cite the reviewer’s papers.